# Deformation Robust Roto-Scale-Translation Equivariant CNNs

**L. Mars Gao**                                                                           *marsgao@uw.edu*
*Paul G. Allen School of Computer Science & Engineering*
*University of Washington*
*Seattle, WA 98195, USA*

**Guang Lin**                                                                            *guanglin@purdue.edu*
*Department of Mathematics and School of Mechanical Engineering*
*Purdue University*
*West Lafayette, IN 47907, USA*

**Wei Zhu**                                                                             *weizhu@umass.edu*
*Department of Mathematics and Statistics*
*University of Massachusetts Amherst*
*Amherst, MA 01003, USA*

**Reviewed on OpenReview:** *https://openreview.net/forum?id=yVkpxs77cD*

## Abstract

Incorporating group symmetry directly into the learning process has proved to be an effective guideline for model design. By producing features that are guaranteed to transform covariantly to the group actions on the inputs, group-equivariant convolutional neural networks (G-CNNs) achieve significantly improved generalization performance in learning tasks with intrinsic symmetry. General theory and practical implementation of G-CNNs have been studied for planar images under either rotation or scaling transformation, but only individually. We present, in this paper, a roto-scale-translation equivariant CNN ($\mathcal{RST}$-CNN), that is guaranteed to achieve equivariance jointly over these three groups via coupled group convolutions. Moreover, as symmetry transformations in reality are rarely perfect and typically subject to input deformation, we provide a stability analysis of the equivariance of representation to input distortion, which motivates the truncated expansion of the convolutional filters under (pre-fixed) low-frequency spatial modes. The resulting model provably achieves deformation-robust $\mathcal{RST}$ equivariance, i.e., the $\mathcal{RST}$ symmetry is still "approximately" preserved when the transformation is "contaminated" by a nuisance data deformation, a property that is especially important for out-of-distribution generalization. Numerical experiments on MNIST, Fashion-MNIST, and STL-10 demonstrate that the proposed model yields remarkable gains over prior arts, especially in the small data regime where both rotation and scaling variations are present within the data.

## 1 Introduction

Symmetry is ubiquitous in machine learning. For instance, in image classification, the class label of an image remains the same when the image is spatially translated. Convolutional neural networks (CNNs) through spatial weight sharing achieve built-in *translation-equivariance*, i.e., a shift of the input leads to a corresponding shift of the output, a property that improves the generalization performance and sample complexity of the model for computer vision tasks with translation symmetry, such as image classification (Krizhevsky et al., 2012), object detection (Ren et al., 2015), and segmentation (Long et al., 2015; Ronneberger et al., 2015).

Inspired by the standard CNNs, researchers in recent years have developed both theoretical foundations and practical implementations of *group equivariant CNNs* (G-CNNs), i.e., generalized CNN models that guarantee a desired transformation on layer-wise features under a given input group transformation, for signals defined on Euclidean spaces (Cohen & Welling, 2016; 2017; Worrall et al., 2017; Weiler et al., 2018b;a; Worrall & Welling, 2019; Sosnovik et al., 2020; Cheng et al., 2019; Zhu et al., 2019; Weiler & Cesa, 2019; Hoogeboom et al., 2018; Zhou et al., 2017; Worrall & Brostow, 2018; Kanazawa et al., 2014; Xu et al., 2014; Marcos et al., 2018), manifolds (Cohen et al., 2018; 2019; Kondor et al., 2018; Defferrard et al., 2020), point clouds (Thomas et al., 2018; Chen et al., 2021; Zhao et al., 2020), and graphs (Kondor, 2018; Anderson et al., 2019; Keriven & Peyré, 2019). In particular, equivariant CNNs for *either* rotation (Weiler & Cesa, 2019; Cheng et al., 2019; Hoogeboom et al., 2018; Worrall et al., 2017; Zhou et al., 2017; Marcos et al., 2017; Weiler et al., 2018b) *or* scaling (Kanazawa et al., 2014; Marcos et al., 2018; Xu et al., 2014; Worrall & Welling, 2019; Sosnovik et al., 2020; Zhu et al., 2019; Sosnovik et al., 2021) transforms on 2D inputs have been well studied *separately*, and their advantage has been empirically verified in settings where the data have rich variance in either rotation or scale *individually*.

However, for many vision tasks, it is beneficial for a model to *simultaneously* incorporate translation, rotation, and scaling symmetry directly into its representation. For example, a self-driving vehicle is required to recognize and locate pedestrians, objects, and road signs under random translation (e.g., moving pedestrians), rotation (e.g., tilted road signs), and scaling (e.g., close and distant objects) (Bojarski et al., 2016). Moreover, in realistic settings, symmetry transformations are rarely perfect; for instance, a tilted stop sign located faraway can be modeled in reality as if it were transformed through a sequence of translation, rotation and scaling following a local data *deformation*, which results from (unavoidable) changing view angle and/or digitization. It is thus crucial to design roto-scale-translation equivariant CNNs ($\mathcal{RST}$-CNNs) with provably robust equivariant representation such that the $\mathcal{RST}$ symmetry is still "approximately" preserved when the transformation is "contaminated" by a nuisance data deformation. Such deformation robustness is especially important for out-of-distribution generalization. However, the design of $\mathcal{RST}$-equivariant CNNs with theoretically guaranteed deformation robustness is challenging both in theory and in practice due to the intertwined convolution on the non-compact $\mathcal{RST}$ group with infinite Haar measure.

The purpose of this paper is to address both the theoretical and practical aspects of constructing deformation robust $\mathcal{RST}$-CNNs, which, to the best of our knowledge, have not been jointly studied in the computer vision community. Specifically, our contribution is three-fold:

1. We propose roto-scale-translation equivariant CNNs with joint convolutions over the space $\mathbb{R}^2$, the rotation group $SO(2)$, and the scaling group $\mathcal{S}$, which is shown to be sufficient and necessary for equivariance with respect to the regular representation of the group $\mathcal{RST}$.

2. We provide a stability analysis of the proposed model, guaranteeing its ability to achieve equivariant representations that are robust to nuisance data deformation.

3. Numerical experiments are conducted to demonstrate the superior (both in-distribution and out-of-distribution) generalization performance of our proposed model for vision tasks with intrinsic $\mathcal{RST}$ symmetry, especially in the small data regime.

## 2    Related Works

**Group-equivariant CNNs (G-CNNs).** Since its introduction by Cohen & Welling (2016), a variety of works on G-CNNs have been conducted that consistently demonstrate the benefits of bringing equivariance prior into network designs. Based on the idea proposed in (Cohen & Welling, 2016) for discrete symmetry groups, G-CNNs with group convolutions which achieve equivariance under regular representations of the group have been studied for the 2D (and 3D) roto-translation group $SE(2)$ (and $SE(3)$) (Weiler & Cesa, 2019; Cheng et al., 2019; Hoogeboom et al., 2018; Worrall et al., 2017; Zhou et al., 2017; Marcos et al., 2017; Weiler et al., 2018b; Worrall & Brostow, 2018), scaling-translation group (Kanazawa et al., 2014; Marcos et al., 2018; Xu et al., 2014; Worrall & Welling, 2019; Sosnovik et al., 2020; Zhu et al., 2019; Sosnovik et al., 2021), rotation $SO(3)$ on the sphere (Cohen et al., 2018; Kondor et al., 2018; Defferrard et al., 2020), and permutation on

graphs (Kondor, 2018; Anderson et al., 2019; Keriven & Peyré, 2019). Polar transformer networks (Esteves et al., 2017) generalizes group-equivariance to rotation and dilation. B-spline CNNs (Bekkers, 2020) and LieConv (Finzi et al., 2020) generalize group convolutions to arbitrary Lie groups on generic spatial data, albeit achieving slightly inferior performance compared to G-CNNs specialized for Euclidean inputs (Finzi et al., 2020). Steerable CNNs further generalize the network design to realize equivariance under induced representations of the symmetry group (Cohen & Welling, 2017; Weiler et al., 2018a; Weiler & Cesa, 2019; Cohen et al., 2019), and the general theory has been summarized by Cohen et al. (2019) for homogeneous spaces.

**Representation robustness to input deformations.** Input deformations typically introduce noticeable yet uninformative variability within the data. Models that are robust to data deformation are thus favorable for many vision applications. The scattering transform network (Bruna & Mallat, 2013; Mallat, 2010; 2012), a multilayer feature encoder defined by average pooling of wavelet modulus coefficients, has been proved to be stable to both input noise and nuisance deformation. Using group convolutions, scattering transform has also been extended in (Oyallon & Mallat, 2015; Sifre & Mallat, 2013) to produce rotation/translation-invariant features. Despite being a pioneering mathematical model, the scattering network uses pre-fixed wavelet transforms in the model, and is thus non-adaptive to the data. Stability and invariance have also been studied in (Bietti & Mairal, 2017; 2019) for convolutional kernel networks (Mairal, 2016; Mairal et al., 2014). DCFNet (Qiu et al., 2018) combines the regularity of a pre-fixed filter basis and the trainability of the expansion coefficients, achieving both representation stability and data adaptivity. The idea is later adopted in (Cheng et al., 2019) and (Zhu et al., 2019) in building deformation robust models that are equivariant to either rotation or scaling individually.

Despite the growing body of literature in G-CNNs, to the best of our knowledge, no G-CNNs have been specifically designed to *simultaneously* achieve roto-scale-translation equivariance. More importantly, no stability analysis has been conducted to quantify and promote robustness of such equivariant model to nuisance input deformation.

## 3  Roto-Scale-Translation Equivariant CNNs

We first explain, in Section 3.1 , the definition of groups, group representations, and group-equivariance, which serves as a background for constructing $\mathcal{RST}$-CNNs in Section 3.2.

### 3.1  Preliminaries

**Group.** A group is a set $G$ equipped with a binary operator $\cdot : G \times G \to G$, satisfying associativity and the existence of an identity $e$ as well as an inverse element $g^{-1}$ for all $g \in G$. In this paper, we consider in particular the roto-scale-translation group $\mathcal{RST} = (SO(2) \times \mathbb{R}) \ltimes \mathbb{R}^2 = \{(\eta, \beta, v) : \eta \in [0, 2\pi], \beta \in \mathbb{R}, v \in \mathbb{R}^2\}$, with the group multiplication

$$(\eta, \beta, v) \cdot (\theta, \alpha, u) = (\theta + \eta, \alpha + \beta, v + R_\eta 2^\beta u), \tag{1}$$

where $R_\eta u$ is a counterclockwise rotation (around the origin) by angle $\eta$ applied to a point $u \in \mathbb{R}^2$.

**Group action and representation.** Given a group $G$ and a set $\mathcal{X}$, $D_g : \mathcal{X} \to \mathcal{X}$ is called a *G-action* on $\mathcal{X}$ if $D_g$ is invertible for all $g \in G$, and $D_{g_1} \circ D_{g_2} = D_{g_1 \cdot g_2}$, $\forall g_1, g_2 \in G$, where $\circ$ denotes map composition. A *G*-action $D_g$ is called a *G-representation* if $\mathcal{X}$ is further assumed to be a vector space and $D_g$ is linear for all $g \in G$. In particular, given an input RGB image $x^{(0)}(u, \lambda)$ modeled in the continuous setting (i.e., $x^{(0)}$ is the intensity of the RGB color channel $\lambda \in \{1, 2, 3\}$ at the pixel location $u \in \mathbb{R}^2$), a roto-scale-translation transformation on the image $x^{(0)}(u, \lambda)$ can be understood as an $\mathcal{RST}$-action (representation) $D_g^{(0)} = D_{\eta, \beta, v}^{(0)}$ acting on the input $x^{(0)}$:

$$[D_{\eta, \beta, v}^{(0)} x^{(0)}](u, \lambda) = x^{(0)} \left( R_{-\eta} 2^{-\beta}(u - v), \lambda \right), \tag{2}$$

i.e., the transformed image $[D_{\eta, \beta, v} x^{(0)}]$ is obtained through an $\eta$ rotation, $2^\beta$ scaling, and $v$ translation.

**Group Equivariance.** Let $f : \mathcal{X} \to \mathcal{Y}$ be a map between $\mathcal{X}$ and $\mathcal{Y}$, and $D_g^{\mathcal{X}}, D_g^{\mathcal{Y}}$ be $G$-actions on $\mathcal{X}$ and $\mathcal{Y}$ respectively. The map $f$ is said to be $G$-equivariant if

$$f(D_g^{\mathcal{X}} x) = D_g^{\mathcal{Y}}(f(x)), \quad \forall\, g \in G,\ x \in \mathcal{X}. \tag{3}$$

A special case of (3) is $G$-invariance, when $D_g^{\mathcal{Y}}$ is set to $\mathrm{Id}_{\mathcal{Y}}$, the identity map on $\mathcal{Y}$. For vision tasks where the output $y \in \mathcal{Y}$ is known a priori to transform covariantly through $D_g^{\mathcal{Y}}$ to a $D_g^{\mathcal{X}}$ transformed input, e.g., the class label $y$ remains identical for a rotated/rescaled/shifted input $x \in \mathcal{X}$, it is beneficial to consider only $G$-equivariant models $f$ to reduce the statistical error of the learning method for improved generalization.

### 3.2 Equivariant Architecture

Since the composition of equivariant maps remains equivariant, to construct an $L$-layer $\mathcal{RST}$-CNN, we only need to specify the $\mathcal{RST}$-action $D_{\eta,\beta,v}^{(l)}$ on each feature space $\mathcal{X}^{(l)}$, $0 \le l \le L$, and require the layer-wise mapping to be equivariant:

$$x^{(l)}[D_{\eta,\beta,v}^{(l-1)} x^{(l-1)}] = D_{\eta,\beta,v}^{(l)} x^{(l)}[x^{(l-1)}],\ \forall l \ge 1, \tag{4}$$

where we slightly abuse the notation $x^{(l)}[x^{(l-1)}]$ to denote the $l$-th layer output given the $(l-1)$-th layer feature $x^{(l-1)}$. In particular, we define $D_{\eta,\beta,v}^{(0)}$ on the input as in (2); for the hidden layers $1 \le l \le L$, we let $\mathcal{X}^{(l)}$ be the feature space consisting of features in the form of $x^{(l)}(u,\theta,\alpha,\lambda)$, where $u \in \mathbb{R}^2$ is the spatial position, $\theta \in [0,2\pi]$ is the rotation index, $\alpha \in \mathbb{R}$ is the scale index, $\lambda \in [M_l] := \{1,\ldots,M_l\}$ corresponds to the unstructured channels (similar to the RGB channels of the input), and we define the action $D_{\eta,\beta,v}^{(l)}$ on $\mathcal{X}^{(l)}$ as

$$[D_{\eta,\beta,v}^{(l)} x^{(l)}](u,\theta,\alpha,\lambda) = x^{(l)}\left(R_{-\eta} 2^{-\beta}(u-v), \theta-\eta, \alpha-\beta, \lambda\right),\ \forall l \ge 1. \tag{5}$$

We note that (5) corresponds to the *regular* representation of $\mathcal{RST}$ on $\mathcal{X}^{(l)}$ (Cohen et al., 2019), which is adopted in this work as its ability to encode any function on the group $\mathcal{RST}$ leads typically to better model expressiveness and stronger generalization performance (Weiler & Cesa, 2019). The following proposition outlines the general network architecture to achieve $\mathcal{RST}$-equivariance under the representations $D_{\eta,\beta,v}^{(l)}$ (2) (5).

**Proposition 1.** *An $L$-layer feedforward neural network is $\mathcal{RST}$-equivariant under the representations (2) (5) if and only if the layer-wise operations are defined as (6) and (7):*

$$x^{(1)}[x^{(0)}](u,\theta,\alpha,\lambda) = \sigma\left[\sum_{\lambda'} \int_{\mathbb{R}^2} x^{(0)}(u+u',\lambda') \cdot 2^{-2\alpha} W_{\lambda',\lambda}^{(1)}\left(2^{-\alpha} R_{-\theta} u'\right) du' + b^{(1)}(\lambda)\right], \tag{6}$$

$$x^{(l)}[x^{(l-1)}](u,\theta,\alpha,\lambda)$$
$$= \sigma\left[\sum_{\lambda'} \int_{\mathbb{R}^2} \int_{S^1} \int_{\mathbb{R}} x^{(l-1)}(u+u',\theta+\theta',\alpha+\alpha',\lambda') \cdot 2^{-2\alpha} W_{\lambda',\lambda}^{(l)}\left(2^{-\alpha} R_{-\theta} u',\theta',\alpha'\right) d\alpha' d\theta' du' + b^{(l)}(\lambda)\right], \tag{7}$$

*where $\sigma : \mathbb{R} \to \mathbb{R}$ is a pointwise nonlinearity, $W_{\lambda',\lambda}^{(1)}(u)$ is the spatial convolutional filter in the first layer with output channel $\lambda$ and input channel $\lambda'$, $W_{\lambda',\lambda}^{(l)}(u,\theta,\alpha)$ is the $\mathcal{RST}$ joint convolutional filter for layer $l > 1$, and $\int_{S^1} f(\alpha) d\alpha$ denotes the normalized $S^1$ integral $\frac{1}{2\pi} \int_0^{2\pi} f(\alpha) d\alpha$.*

We note that the joint-convolution (7) is the group convolution over $\mathcal{RST}$ (whose subgroup $SO(2) \times \mathbb{R}$ is a non-compact Lie group acting on $\mathbb{R}^2$), which is known to achieve equivariance under regular representations (5) (Cohen et al., 2019). We provide an elementary proof of Proposition 1 in the appendix for completeness.

## 4 Robust Equivariance to Input Deformation

Proposition 1 details the network architecture to achieve $\mathcal{RST}$-equivariance for images modeled on the *continuous* domain $\mathbb{R}^2$ undergoing a "*perfect*" $\mathcal{RST}$-transformation (2). However, in practice, symmetry

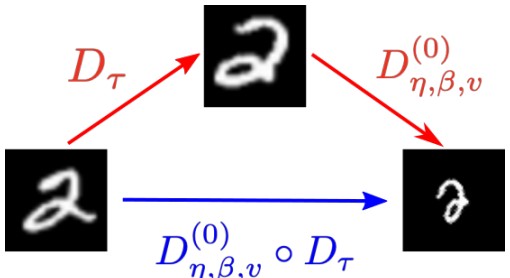

Figure 1: $\mathcal{RST}$-transformations in reality, e.g., the blue arrow, are rarely perfect, and they can be modeled as a composition of input deformation $D_\tau$ [cf. (12)] and an exact $\mathcal{RST}$-transformation $D_{\eta,\beta,v}^{(0)}$.

transformations are rarely perfect, as they typically suffer from numerous source of input deformation coming from, for instance, unstable camera position, change of weather, as well as practical issues such as numerical discretization and truncation (see, for example, Figure 1.) We explain, in this section, how to quantify and improve the representation stability of the model such that it stays "approximately" $\mathcal{RST}$-equivariant even if the input transformation is "contaminated" by minute local distortion (see, for instance, Figure 2.)

## 4.1 Decomposition of Convolutional Filters

In order to quantify the deformation stability of representation equivariance, motivated by (Qiu et al., 2018; Cheng et al., 2019; Zhu et al., 2019), we leverage the geometry of the group $\mathcal{RST}$ and decompose the convolutional filters $W_{\lambda,\lambda'}^{(l)}(u,\theta,\alpha)$ under the separable product of three orthogonal function bases, $\{\psi_k(u)\}_k$, $\{\varphi_m(\theta)\}_m$, and $\{\xi_n(\alpha)\}_n$. In particular, we choose $\{\varphi_m\}_m$ as the Fourier basis on $S^1$, and set $\{\psi_k\}_k$ and $\{\xi_n\}_n$ to be the eigenfunctions of the Dirichlet Laplacian over, respectively, the unit disk $D \subset \mathbb{R}^2$ and the interval $I_\alpha = [-1,1]$, i.e.,

$$\begin{cases} \Delta\psi_k = -\mu_k\psi_k \ \text{ in } D, \\ \qquad \psi_k = 0 \ \text{ on } \ \partial D, \end{cases} \quad \begin{cases} \xi_n'' = -\nu_m\xi_n \ \text{ in } I_\alpha \\ \xi_n(\pm 1) = 0, \end{cases} \tag{8}$$

where $\mu_k$ and $\psi_n$ are the corresponding eigenvalues.

**Remark 1.** *One has flexibility in choosing the spatial function basis $\{\psi_k\}_k$. We consider mainly, in this work, the rotation-steerable Fourier-Bessel (FB) basis (Abramowitz & Stegun, 1965) defined in (8), as the spatial regularity of its low-frequency modes leads to robust equivariance to input deformation in an $\mathcal{RST}$-CNN, which will be shown in Theorem 1. One can also choose $\{\psi_k\}_k$ to be the eigenfunctions of Dirichlet Laplacian over the cell $[-1,1]^2$ (Zhu et al., 2019), i.e., the separable product of the solutions to the 1D Dirichlet Sturm–Liouville problem, which leads to a similar stability analysis. We denote such basis the Sturm-Liouvielle (SL) basis, and its efficacy will be compared to FB basis in Section 6.*

Since spatial pooling can be modeled as rescaling the convolutional filters in space, we assume the filters $W_{\lambda',\lambda}^{(l)}$ are compactly supported on a rescaled domain as follows

$$W_{\lambda',\lambda}^{(1)} \in C_c(2^{j_1}D), \ W_{\lambda',\lambda}^{(l)} \in C_c(2^{j_l}D \times S^1 \times I_\alpha), \tag{9}$$

where $j_l < j_{l+1}$ models a sequence of filters with decreasing size. Let $\psi_{j,k}(u) = 2^{-2j}\psi_k(2^{-j}u)$ be the rescaled spatial basis function, and we can decompose $W_{\lambda',\lambda}^{(l)}$ under $\{\psi_{j_l,k}\}_k$, $\{\varphi_m\}_m$, $\{\xi_n\}_n$ into

$$W_{\lambda',\lambda}^{(1)}(u) = \sum_k a_{\lambda',\lambda}^{(1)}(k)\psi_{j_1,k}(u), \quad W_{\lambda',\lambda}^{(l)}(u,\theta,\alpha) = \sum_{k,m,n} a_{\lambda',\lambda}^{(l)}(k,m,n)\psi_{j_1,k}(u)\varphi_m(\theta)\xi_n(\alpha), \tag{10}$$

where $a_{\lambda',\lambda}^{(l)}$ are the expansion coefficients of the filters. In practice, the filter expansion (10) are truncated to only low-frequency components of the separable basis, which reduces the computation and memory cost of the model. In addition, we will explain in Section 4.2 the effect of the truncated filter expansion on the stability of the $\mathcal{RST}$-equivariant representation to input deformation.

## 4.2 Stability under Input Deformation

First, in order to gauge the distance between different inputs and features, we define the layer-wise feature norm as

$$\left\|x^{(0)}\right\|^2 = \frac{1}{M_0}\sum_{\lambda=1}^{M_0}\int_{\mathbb{R}^2}|x^{(0)}(u,\lambda)|^2 du, \quad \left\|x^{(l)}\right\|^2 = \sup_{\alpha}\frac{1}{M_l}\sum_{\lambda=1}^{M_l}\iint_{S^1\times\mathbb{R}^2}|x^{(l)}(u,\theta,\alpha,\lambda)|^2 dud\theta, \tag{11}$$

for $l \geq 1$, i.e., the norm is a combination of an $L^2$-norm over the roto-translation group $SE(2)\cong S^1\times\mathbb{R}^2$ and an $L^\infty$-norm over the scaling group $\mathcal{S}\cong\mathbb{R}$. We point out the importance of the $L^\infty$-norm in $\mathcal{S}$, as signals after the coupled convolution (7) generally do not vanish as $\alpha\to-\infty$.

We next define the spatial deformation of an input image. Given a $C^2$ vector field $\tau:\mathbb{R}^2\to\mathbb{R}^2$, the spatial deformation $D_\tau$ on $x^{(0)}$ is defined as

$$D_\tau x^{(0)}(u,\lambda) = x^{(0)}(\rho(u),\lambda), \tag{12}$$

where $\rho(u) = u - \tau(u)$. Thus $\tau(u)$ is understood as the local image distortion (at pixel location $u$), and $D_\tau$ is the identity map if $\tau(u)\equiv 0$, i.e., not input distortion.

The deformation stability of the equivariant representation can be quantified in terms of (11) after we make the following three mild assumptions on the model and the input distortion $D_\tau$:

**(A1):** The nonlinearity $\sigma:\mathbb{R}\to\mathbb{R}$ is non-expansive, i.e., $|\sigma(x)-\sigma(y)|\leq|x-y|$, $\forall x,y\in\mathbb{R}$. For instance, the rectified linear unit (ReLU) satisfies this assumption.

**(A2):** The convolutional filters are bounded in the following sense: $A_l\leq 1, \forall l\geq 1$, where

$$A_1 := \pi\max\left\{\sup_\lambda\sum_{\lambda'}\|a_{\lambda',\lambda}^{(1)}\|_{\mathrm{FB}}, \frac{M_0}{M_1}\sup_{\lambda'}\sum_{\lambda'}\|a_{\lambda',\lambda}^{(1)}\|_{\mathrm{FB}}\right\}$$

$$A_l := \pi\max\left\{\sup_\lambda\sum_{\lambda'}\sum_n\|a_{\lambda',\lambda}^{(l)}(\cdot,n)\|_{\mathrm{FB}}, \frac{2M_{l-1}}{M_l}\sum_n\sup_{\lambda'}\sum_\lambda\|a_{\lambda',\lambda}^{(l)}(\cdot,n)\|_{\mathrm{FB}}\right\}, \ \forall l>1, \tag{13}$$

in which the FB-norm $\|\cdot\|_{\mathrm{FB}}$ of a sequence $\{a(k)\}_{k\geq 0}$ and double sequence $\{b(k,m)\}_{k,m\geq 0}$ is the weighted $l_2$-norm defined as

$$\|a\|_{\mathrm{FB}}^2 = \sum_k\mu_k a(k)^2, \ \|b\|_{\mathrm{FB}}^2 = \sum_k\sum_m\mu_k b(k,m)^2, \tag{14}$$

$\mu_k$ being the eigenvalues defined in (8). This technical assumption **(A2)** on the FB-norms of the expansion coefficients, $\|a_{\lambda',\lambda}^{(l)}\|_{\mathrm{FB}}$, ensures the boundedness of the convolutional filters $W_{\lambda',\lambda}^{(l)}(u,\theta,\alpha)$ under various norms (the exact details can be found in Lemma 2 of the appendix), which in turn quantifies the stability of the equivariant representation to input deformation. We point out that the boundedness of $A_l$ can be facilitated by truncating the expansion coefficients to low-frequency components (i.e., small $\mu_k$'s), which is the key idea of our $\mathcal{RST}$-CNN (cf. Remark 3).

**(A3):** The local input distortion is small:

$$|\nabla\tau|_\infty := \sup_u\|\nabla\tau(u)\| < 1/5, \tag{15}$$

where $\|\cdot\|$ is the operator norm.

Theorem 1 below quantifies the deformation stability of the equivariant representation in an $\mathcal{RST}$-CNN under the assumptions **(A1)-(A3)**:

**Theorem 1.** *Let $D_{\eta,\beta,v}^{(l)}$, $0\leq l\leq L$, be the $\mathcal{RST}$ group actions defined in (2)(5), and let $D_\tau$ be a small input deformation define in (12). If an $\mathcal{RST}$-CNN satisfies the assumptions **(A1)-(A3)**, we have, for any $L$,*

$$\left\|x^{(L)}[D_{\eta,\beta,v}^{(0)}\circ D_\tau x^{(0)}] - D_{\eta,\beta,v}^{(L)}x^{(L)}[x^{(0)}]\right\| \leq 2^{\beta+1}\left(4L|\nabla\tau|_\infty + 2^{-j_L}|\tau|_\infty\right)\|x^{(0)}\|. \tag{16}$$

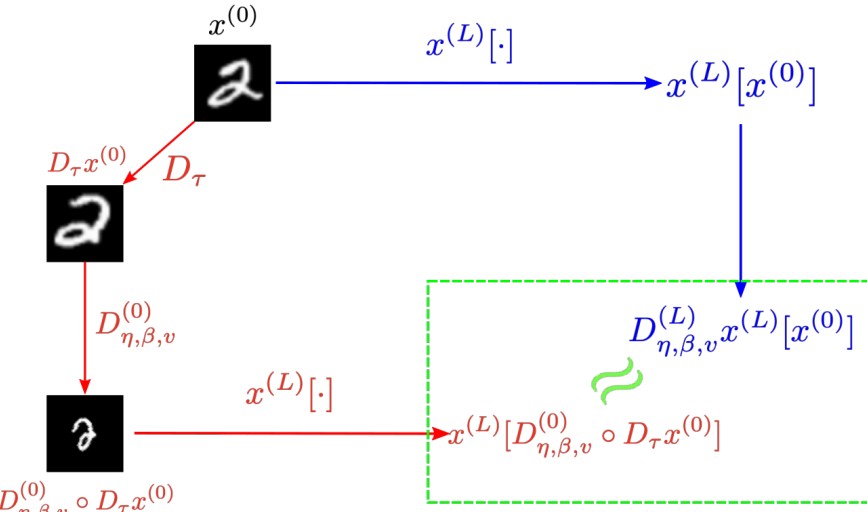

Figure 2: Approximate $\mathcal{RST}$-equivariance in the presence of nuisance input deformation $D_\tau$.

The proof of Theorem 1 is deferred to the Appendix. An important message from Theorem 1 is that as long as **(A1)**-**(A3)** are satisfied, the model stays approximately $\mathcal{RST}$-equivariant, i.e., $x^{(L)}[D_{\eta,\beta,v}^{(0)} \circ D_\tau x^{(0)}] \approx D_{\eta,\beta,v}^{(L)} x^{(L)}[x^{(0)}]$, even with the presence of non-zero (yet small) input deformation $D_\tau$ (see, e.g., Figure 2.)

**Remark 2.** *The fact that the right hand side of* (16) *grows exponentially with $\beta$ is inevitable, as it comes naturally from the definition of the norm* (11)*: if an image is spatially rescaled by $2^\beta$ without correspondingly scaling the color range (i.e., pixel intensity), its $L^2$-norm is enlarged by $2^\beta$.*

**Remark 3.** *According to the definition of the FB-norm* (14)*, the main assumption **(A2)** can be facilitated by truncating the filter expansion* (10) *to include only low-frequency (small $\mu_k$) components. The implementation detail of such truncated filter expansion will be explained in detail in Section 5.*

## 5   Implementation Details

We next discuss the implementation details of the $\mathcal{RST}$-CNN outlined in Proposition 1.

**Discretization.** To implement $\mathcal{RST}$-CNN in practice, we first need to discretize the features $x^{(l)}$ modeled originally under the continuous setting. First, the input signal $x^{(0)}(u, \lambda)$ is discretized on a uniform grid into a 3D array of shape $[M_0, H_0, W_0]$, where $H_0, W_0, M_0$, respectively, are the height, width, and the number of the unstructured channels of the input (e.g., $M_0 = 3$ for RGB images.) For $l \geq 1$, the rotation group $S^1$ is uniformly discretized into $N_r$ points; the scaling group $\mathcal{S} \cong \mathbb{R}^1$, unlike $S^1$, is unbounded, and thus features $x^{(l)}$ are computed and stored only on a truncated scale interval $I = [-T, T] \subset \mathbb{R}$, which is uniformly discretized into $N_s$ points. The feature $x^{(l)}$ is therefore stored as a 5D array of shape $[M_l, N_r, N_s, H_l, W_l]$.

**Filter expansion.** The analysis in Section 4 suggests that robust $\mathcal{RST}$-equivariance is achieved if the convolutional filters are expanded with only the first $K$ low-frequency spatial modes $\{\psi_k\}_{k=1}^K$. More specifically, the first $K$ spatial basis functions as well as their rotated and rescaled versions $\{2^{-2\alpha}\psi_k(2^{-\alpha}R_{-\theta}u')\}_{k,\theta,\alpha,u'}$ are sampled on a uniform grid of size $L \times L$ and stored as an array of size $[K, N_r, N_s, L, L]$, which is fixed during training. The expansion coefficients $a_{\lambda',\lambda}^{(l)}$, on the other hand, are the trainable parameters of the model, which are used together with the fixed basis to linearly expand the filters. The resulting filters $\{2^{-2\alpha}W_{\lambda',\lambda}^{(1)}(2^{-\alpha}R_{-\theta}u')\}_{\lambda',\lambda,\theta,\alpha,u'}$ and $\{2^{-2\alpha}W_{\lambda',\lambda}^{(l)}(2^{-\alpha}R_{-\theta}u', \theta', \alpha')\}_{\lambda',\lambda,\theta,\theta',\alpha,\alpha',u'}$ are stored, respectively, as tensors of size $[M_0, M_1, N_r, N_s, L, L]$ and $[M_{l-1}, M_l, N_r, L_\theta, N_s, L_\alpha, L, L]$, where $L_\alpha$ is the number of grid points sampling the interval $I_\alpha$ in (8) (this is typically smaller than $N_s$, i.e., the number of the grid points discretizing the scale interval $I = [-T, T]$), and $L_\theta$ is the number of grid points sampling $S^1$ on which the integral $\int_{S^1} d\theta$ in (7) is performed.

**Remark 4.** *The number $L_\alpha$ measures the support $I_\alpha$ (8) of the convolutional filters in scale, which corresponds to the amount of "inter-scale" information transfer when performing the convolution over scale $\int_{\mathbb{R}}(\cdots)d\alpha'$ in (7). It is typically chosen to be a small number (e.g., 1 or 2) to avoid the "boundary leakage effect" (Worrall & Welling, 2019; Sosnovik et al., 2020; Zhu et al., 2019), as one needs to pad unknown values beyond the truncated scale channel $[-T, T]$ during convolution (7) when $L_\alpha > 1$. The number $L_\theta$, on the other hand, corresponds to the "inter-rotation" information transfer when performing the convolution over the rotation group $\int_{S^1}(\cdots)d\theta'$ in (7); it does not have to be small since periodic-padding of known values is adopted when conducting integrals on $S^1$ with no "boundary leakage effect". We only require $L_\theta$ to divide $N_r$ such that $\int_{S^1}(\cdots)d\theta'$ is computed on a (potentially) coarser grid (of size $L_\theta$) compared to the finer grid (of size $N_r$) on which we discretize the rotation channel of the feature $x^{(l)}$.*

**Discrete convolution.** After generating, in the previous step, the discrete joint convolutional filters together with their rotated and rescaled versions, the continuous convolutions in Proposition 1 can be efficiently implemented using regular 2D discrete convolutions.

More specifically, let $x^{(0)}(u, \lambda)$ be an input image of shape $[M_0, H_0, W_0]$. A total of $N_r \times N_s$ discrete 2D convolutions with the rotated and rescaled filters $\{2^{-2\alpha}\psi_k(2^{-\alpha}R_{-\theta}u')\}_{k,\theta,\alpha,u'}$, i.e., replacing the spatial integrals in (7) by summations, are conducted to obtain the first-layer feature $x^{(1)}(u, \theta, \alpha, \lambda)$ of size $[M_1, N_r, N_s, H_1, W_1]$. For the subsequent layers, given a feature $x^{(l-1)}(u, \theta, \alpha, \lambda)$ of shape $[M_{l-1}, N_r, N_s, H_{l-1}, W_{l-1}]$ and the joint filters $F^{(l)} = \{2^{-2\alpha}W^{(l)}_{\lambda',\lambda}(2^{-\alpha}R_{-\theta}u', \theta', \alpha')\}_{\lambda',\lambda,\theta,\alpha,\alpha',u'}$ of size $[M_{l-1}, M_l, N_r, L_\theta, N_s, L_\alpha, L, L]$, the next-layer feature $x^{(l)}$ is computed in the following way: for each $l_\alpha \in [0, L_\alpha - 1]$ and $l_\theta \in [0, L_\theta - 1]$, we shift the signal $x^{(l-1)}$ in the scale channel by $l_\alpha$ and in the rotation channel by $l_\theta N_r / L_\theta$, which is then convolved with the filter $F[:,:,:,l_\theta,:,l_\alpha,:,:]$ (after proper reshaping and combining adjacent dimensions) to produce an output array of shape $[M_l, N_r, N_s, H_l, W_l]$. The $l$-th layer feature map $x^{(l)}(u, \theta, \alpha, \lambda)$ is then computed as the sum of the $L_\theta \times L_\alpha$ tensors obtained by iterating over $l_\theta \in [0, L_\theta - 1]$ and $l_\alpha \in [0, L_\alpha - 1]$.

**Group pooling.** For learning tasks where the outputs are supposed to remain unchanged to $\mathcal{RST}$-transformed inputs, e.g., image classification, a max-pooling over the entire group $\mathcal{RST}$ is performed on the last-layer feature $x^{(L)}(u, \theta, \alpha, \lambda)$ of shape $[M_L, N_r, N_s, H_L, M_L]$ to produce an $\mathcal{RST}$-invariant 1D output of length $M_L$. We only perform the $\mathcal{RST}$ group-pooling in the last layer without explicit mention.

## 6 Numerical Experiments

We conduct numerical experiments, in this section, to demonstrate:

- The proposed model indeed achieves robust $\mathcal{RST}$-equivariance under realistic settings.

- $\mathcal{RST}$-CNN yields remarkable gains over prior arts in vision tasks with intrinsic $\mathcal{RST}$-symmetry, especially in the small data regime.

Software implementation of the experiments is included in the supplementary materials.

### 6.1 Data Sets and Models

We conduct the experiments on the Rotated-and-Scaled MNIST (RS-MNIST), Rotated-and-Scaled Fashion-MNIST (RS-Fashion), SIM2MNIST (Esteves et al., 2017), as well as the STL-10 data sets (Coates et al., 2011b).

RS-MNIST and RS-Fashion are constructed through randomly rotating (by an angle uniformly distributed on $[0, 2\pi]$) as well as rescaling (by a uniformly random factor from $[0.3, 1]$) the original MNIST (LeCun et al., 1998) and Fashion-MNIST (Xiao et al., 2017a) images. The transformed images are zero-padded back to a size of $28 \times 28$. We upsize the image to $56 \times 56$ for better comparison of the models. In addition, we also consider the noise-introduced SIM2MNIST data set, obtained from randomly rotating (uniformly on $[0, 2\pi]$), rescaling (uniformly on $[1, 2.4]$), and translating the MNIST images before upsampling to a spatial dimension of $96 \times 96$. The resulting SIM2MNIST data set is split into 10K, 5K, 50K samples for training, validation, and testing respectively.

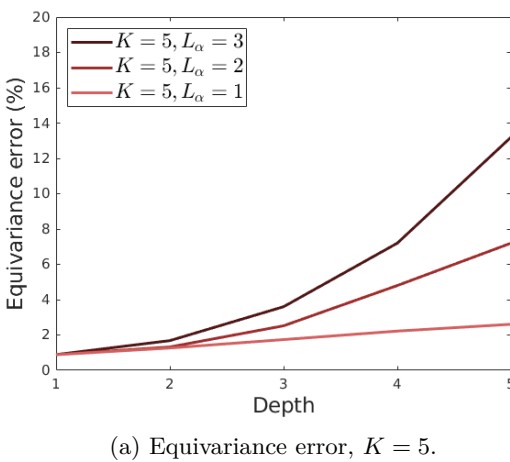
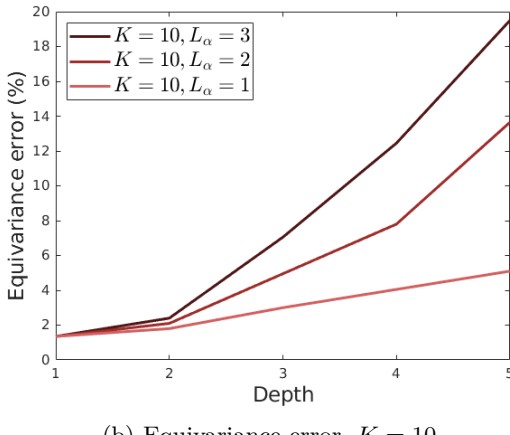

(a) Equivariance error, $K = 5$.
(b) Equivariance error, $K = 10$.

Figure 3: Layer-wise equivariance error (17) of the $\mathcal{RST}$-CNN. Convolutional filters with varying number $L_\alpha$ of "inter-scale" channels are constructed from $K$ low-frequency spatial modes $\{\psi_k\}_{k=1}^K$. The equivariance error is smaller when $K$ decreases, verifying our theoretical analysis (Theorem 1) that truncated basis expansion to low-frequency modes improves deformation robustness of the equivariant representation.

The STL-10 data set has 5,000 training and 8,000 testing RGB images of size $96 \times 96$ belonging to 10 different classes such as cat, deer, and dog. We use this data set to evaluate different models under both in-distribution (ID) and out-of-distribution (OOD) settings. More specifically, the training set remains unchanged, while the testing set is either unaltered for ID testing, or randomly rotated (by an angle uniformly distributed on $[-\pi/2, \pi/2]$) and rescaled (by a factor uniformly distributed on $[0.8, 1]$) for OOD testing.

We evaluate the performance of the proposed $\mathcal{RST}$-CNN against other models that are equivariant to either roto-translation ($SE(2)$) or scale-translation ($\mathcal{ST}$) of the inputs. The $SE(2)$-equivariant models consider in this section include the Rotation Decomposed Convolutional Filters network (RDCF) (Cheng et al., 2019) and the Rotation Equivariant Steerable Network (RESN) (Weiler et al., 2018b), which is shown to achieve best performance among all $SE(2)$-equivariant CNNs in (Weiler & Cesa, 2019). The $\mathcal{ST}$-equivariant models include the Scale Equivariant Vector Field Network (SEVF) (Marcos et al., 2018), Scale Equivariant Steerable Network (SESN) (Sosnovik et al., 2020), and Scale Decomposed Convolutional Filters network (SDCF) (Zhu et al., 2019).

### 6.2 Equivariance Error

We first measure the $\mathcal{RST}$-equivariance error of our proposed model with the presence of discretization and scale channel truncation. More specifically, we construct a 5-layer $\mathcal{RST}$-CNN with randomly initialized expansion coefficients $a_{\lambda',\lambda}^{(l)}$ truncated to $K = 5$ or $K = 10$ low-frequency spatial (FB) modes $\{\psi_k\}_{k=1}^K$. The scale channel is truncated to $[-1, 1]$, which is uniformly discretized into $N_s = 9$ points; the rotation group $S^1$ is sampled on a uniform grid of size $N_r = 8$. The $\mathcal{RST}$ equivariance error is computed on random RS-MNIST images, and measured in a relative $L^2$ sense at the scale $\alpha = 0$ and rotation $\theta = 0$, with the $\mathcal{RST}$-action corresponding to the group element $(\eta, \beta, v) = (-\pi/2, -0.5, 0)$, i.e.,

$$\frac{\left\| \left( x^{(l)}[D_{\eta,\beta,v} x^{(0)}] - D_{\eta,\beta,v} x^{(l)}[x^{(0)}] \right)(\cdot, \alpha, \theta) \right\|_{L^2}}{\left\| D_{\eta,\beta,v} x^{(l)}[x^{(0)}](\cdot, \alpha, \theta) \right\|_{L^2}} \tag{17}$$

We fix $L_\theta$, i.e., the number of the "inter-rotation" channels corresponding to the "coarser" grid of $S^1$ for discrete $S^1$ integration, to $L_\theta = 4$, and examine the equivariance error induced by the "boundary leakage effect" with different numbers $L_\alpha$ of the "inter-scale" channels [cf. Remark 4].

Figure 3 displays the equivariance error (17) of the $\mathcal{RST}$-CNN at different layers $l \in \{1, \cdots, 5\}$ with varying $L_\alpha \in \{1, 2, 3\}$. It can be observed that the equivariance error is inevitable due to numerical discretization and

| Models | RS-MNIST test accuracy (%) | | RS-MNIST+ test accuracy (%) | |
|---|---|---|---|---|
| | $N_{\mathrm{tr}} = 2,000$ | $N_{\mathrm{tr}} = 5,000$ | $N_{\mathrm{tr}} = 2,000$ | $N_{\mathrm{tr}} = 5,000$ |
| CNN | $80.63 \pm 0.11$ | $87.41 \pm 0.32$ | $89.52 \pm 0.21$ | $93.08 \pm 0.04$ |
| RESN | $85.71 \pm 0.18$ | $89.69 \pm 0.40$ | $90.86 \pm 0.20$ | $93.78 \pm 0.35$ |
| RESN+ | $87.96 \pm 0.05$ | $92.29 \pm 0.10$ | $92.20 \pm 0.09$ | $95.32 \pm 0.09$ |
| RDCF | $86.79 \pm 0.12$ | $90.46 \pm 0.33$ | $90.40 \pm 0.31$ | $94.84 \pm 0.15$ |
| RDCF+ | $87.68 \pm 0.52$ | $91.74 \pm 0.37$ | $92.49 \pm 0.28$ | $95.81 \pm 0.11$ |
| SEVF | $85.76 \pm 0.38$ | $90.29 \pm 0.37$ | $89.97 \pm 0.42$ | $93.47 \pm 0.18$ |
| SESN | $84.58 \pm 0.29$ | $90.19 \pm 0.39$ | $90.33 \pm 0.30$ | $93.40 \pm 0.29$ |
| SDCF | $85.62 \pm 0.51$ | $90.40 \pm 0.09$ | $90.14 \pm 0.16$ | $93.47 \pm 0.05$ |
| $\mathcal{RST}$-CNN(FB) | $89.16 \pm 0.32$ | $93.19 \pm 0.29$ | $92.58 \pm 0.35$ | $96.33 \pm 0.26$ |
| $\mathcal{RST}$-CNN(SL) | $88.97 \pm 0.17$ | $93.03 \pm 0.20$ | $92.30 \pm 0.19$ | $96.04 \pm 0.11$ |
| $\mathcal{RST}$-CNN+(FB) | $89.53 \pm 0.27$ | $93.40 \pm 0.26$ | $\mathbf{93.99 \pm 0.07}$ | $96.53 \pm 0.25$ |
| $\mathcal{RST}$-CNN+(SL) | $\mathbf{90.26 \pm 0.37}$ | $\mathbf{93.59 \pm 0.06}$ | $93.82 \pm 0.25$ | $\mathbf{96.76 \pm 0.11}$ |

Table 1: Classification accuracy on the RS-MNIST data set. Models are trained on $N_{\mathrm{tr}} = 2K$ or $5K$ images with spatial resolution $56 \times 56$. A plus sign "+" on the data, i.e., RS-MNIST+, is used to denote the presence of data augmentation during training. A plus sign "+" on the model, e.g., RDCF+, denotes a larger network with more "inter-rotation" correlation $L_\theta = 4$ [cf. Section 5]. The mean $\pm$ std of the test accuracy over five independent trials are reported.

truncation as the model goes deeper. However, it can be mitigated by choosing a small $L_\alpha$, i.e., less "inter-scale" information transfer, to avoid the "boundary leakage effect", or expanding the filters with a small number $K$ of low-frequency spatial components $\{\psi_k\}_{k=1}^K$, supporting our theoretical analysis Theorem 1. Due to this finding, we will consider in the following experiments $\mathcal{RST}$-CNNs with $L_\alpha = 1$, which has the additional benefit of better model scalability due to the reduced scale channel convolution.

## 6.3    Image Classification

We next demonstrate the superior performance of the proposed $\mathcal{RST}$-CNN in image classification under settings where a large variation of rotation and scale is present in the test and/or the training data.

### 6.3.1    RS-MNIST, RS-Fashion and SIM2MNIST

We first benchmark the performance of different models on the RS-MNIST, and RS-Fashion data sets. We generate five independent realizations of the rotated and rescaled data [cf. Secion 6.1], which are split into $N_{\mathrm{tr}} = 5,000$ or $2,000$ images for training, $2,000$ images for validation, and $50,000$ images for testing.

For fair comparison among different models, we use a benchmark CNN with three convolutional and two fully-connected layers as a baseline. Each hidden layer (i.e., the three convolutional and the first fully-connected layer) is followed by a batch-normalization, and we set the number of output channels of the hidden layers to $[32, 63, 95, 256]$. The size of the convolutional filters is set to $7 \times 7$ for each layer in the CNN. All comparing models (those in Table 1-4 without "+" after the name of the model) are built on the same CNN baseline, and we keep the trainable parameters almost the same ($\sim$500K) by modifying the number of the unstructured channels. For models that are equivariant to rotation (including RESN, RDCF, and $\mathcal{RST}$-CNN), we set the number $N_r$ of rotation channels to $N_r = 8$ [cf. Section 5]; for scale-equivariant models (including SESN, SDCF, and $\mathcal{RST}$-CNN), the number $N_s$ of scale channels is set to $N_s = 4$. In addition, for rotation-equivariant CNNs, we also construct larger models (with the number of trainable parameters$\sim$1.6M) after increasing the "inter-rotation" information transfer [cf. Section 5] by setting $L_\theta = 4$; we attach a "+" symbol to the end of the model name (e.g., RDCF+) to denote such larger models with more "inter-rotation". A group max-pooling is performed only after the final convolutional layer to achieve group-invariant representations for classification. Moreover, for $\mathcal{RST}$-CNN, we consider two different spatial

| Models | RS-Fashion test accuracy (%) | | RS-Fashion+ test accuracy (%) | |
|---|---|---|---|---|
| | $N_{\mathrm{tr}} = 2,000$ | $N_{\mathrm{tr}} = 5,000$ | $N_{\mathrm{tr}} = 2,000$ | $N_{\mathrm{tr}} = 5,000$ |
| CNN | $62.71 \pm 0.37$ | $67.91 \pm 0.28$ | $67.92 \pm 0.12$ | $72.41 \pm 0.46$ |
| RESN | $70.80 \pm 0.41$ | $75.80 \pm 0.11$ | $74.10 \pm 0.46$ | $77.76 \pm 0.16$ |
| RESN+ | $71.59 \pm 0.71$ | $76.32 \pm 0.26$ | $76.80 \pm 0.55$ | $80.89 \pm 0.41$ |
| RDCF | $70.72 \pm 0.10$ | $73.96 \pm 0.19$ | $73.46 \pm 0.10$ | $77.53 \pm 0.11$ |
| RDCF+ | $71.27 \pm 0.34$ | $75.94 \pm 0.35$ | $76.87 \pm 0.57$ | $80.66 \pm 0.48$ |
| SEVF | $66.57 \pm 0.32$ | $71.03 \pm 0.31$ | $68.83 \pm 0.52$ | $73.25 \pm 0.22$ |
| SESN | $66.28 \pm 0.14$ | $72.19 \pm 0.05$ | $69.43 \pm 0.07$ | $75.85 \pm 0.26$ |
| SDCF | $66.29 \pm 0.23$ | $72.24 \pm 0.23$ | $68.40 \pm 0.05$ | $75.11 \pm 0.18$ |
| $\mathcal{RST}$-CNN(FB) | $73.31 \pm 0.16$ | $78.64 \pm 0.60$ | $76.43 \pm 0.59$ | $81.93 \pm 0.04$ |
| $\mathcal{RST}$-CNN(SL) | $72.90 \pm 0.34$ | $78.37 \pm 0.22$ | $76.06 \pm 0.13$ | $80.81 \pm 0.29$ |
| $\mathcal{RST}$-CNN+(FB) | $74.37 \pm 0.08$ | $79.19 \pm 0.36$ | $\mathbf{80.65 \pm 0.31}$ | $\mathbf{84.37 \pm 0.19}$ |
| $\mathcal{RST}$-CNN+(SL) | $\mathbf{74.68 \pm 0.29}$ | $\mathbf{79.81 \pm 0.06}$ | $80.65 \pm 0.46$ | $84.09 \pm 0.09$ |

Table 2: Classification accuracy on the RS-Fashion data set. Models are trained on $N_{\mathrm{tr}} = 2K$ or $5K$ images with spatial resolution $56 \times 56$. A plus sign "+" on the data, i.e., RS-Fashion+, is used to denote the presence of data augmentation during training. A plus sign "+" on the model, e.g., RDCF+, denotes a larger network with more "inter-rotation" correlation $L_\theta = 4$ [cf. Section 5]. The mean $\pm$ std of the test accuracy over five independent trials are reported.

| Models | CNN | RDCF | RESN | SEVF | SESN | SDCF | $\mathcal{RST}$-CNN | $\mathcal{RST}$-CNN+ |
|---|---|---|---|---|---|---|---|---|
| Accuracy (%) | $86.6 \pm 0.14$ | $89.3 \pm 0.37$ | $89.2 \pm 0.36$ | $87.31 \pm 0.05$ | $87.86 \pm 0.31$ | $88.1 \pm 0.62$ | $93.34 \pm 0.13$ | $\mathbf{94.81 \pm 0.12}$ |

Table 3: Classification accuracy on the SIM2MNIST data set. The mean $\pm$ std of the test accuracy over three independent trials are reported.

function basis for filter expansion, namely the Fourier-Bessel (FB) basis, and Sturm-Liouville (SL) basis [cf. Remark 1].

We use the Adam optimizer (Kingma & Ba, 2014) to train all models for 60 epochs with the batch size set to 128. We set the initial learning rate to 0.01, which is scheduled to decrease tenfold after 30 epochs. We conduct the experiments in 4 different settings, where the number $N_{\mathrm{tr}}$ of training samples is either 2,000 or 5,000, and the models are trained with or without $\mathcal{RST}$ data augmentation.

We report the mean $\pm$ std of the test accuracy after five independent trials in Table 1 and Table 2, where, for example, RS-MNIST (or RS-MNIST+) denotes models are trained on the RS-MNIST data set without (or with) data augmentation. It is clear from Table 1 and Table 2 that $\mathcal{RST}$-CNN has superior generalization capability compared to other models with approximately the same trainable parameters, especially in the small data regime. Furthermore, $\mathcal{RST}$-CNN+ with more inter-rotation correlation (i.e., $L_\theta = 4$) has further improved performance, achieving the best accuracy among all methods. It can also be observed that neither of the two spatial basis functions (i.e., FB and SL) has significant advantage over the other. The preferred choice of the basis for filter expansion could depend on experimental settings including sample size, the number of filters, original data characteristics and the presence (or lack thereof) of data augmentation.

Similarly, adopting the same hyper-parameter setting as the RS-MNIST experiment, we report the test accuracy after three independent trails on the SIM2MNIST data set (Esteves et al., 2017) in Table 3. One can observe that $\mathcal{RST}$-CNN once again outperforms other models with approximately the same number of parameters. In addition, $\mathcal{RST}$-CNN+ with inter-rotation correlation ($L_\theta = 4$) further improves $\mathcal{RST}$-CNN, achieving the best performance among all comparing methods.

| Models | ID Accuracy (%) | OOD Accuracy (%) |
|---|---|---|
| ResNet-16 | $82.66 \pm 0.53$ | $37.63 \pm 1.95$ |
| RESN | $83.84 \pm 0.67$ | $51.28 \pm 2.29$ |
| RDCF | $83.66 \pm 0.57$ | $51.12 \pm 4.21$ |
| SESN | $83.79 \pm 0.24$ | $47.26 \pm 0.63$ |
| SDCF | $83.83 \pm 0.41$ | $43.60 \pm 0.87$ |
| $\mathcal{RST}$-ResNet | $\mathbf{84.08 \pm 0.11}$ | $\mathbf{58.31 \pm 3.62}$ |

Table 4: Test accuracy on the STL-10 data set for both in-distribution (ID) and out-of-distribution (OOD) settings.

### 6.3.2 STL-10

Finally, we use the STL-10 data set as an example to showcase both ID and OOD generalization capacity of the proposed $\mathcal{RST}$-ResNet. As explained in Section 6.1, the training set remains unaltered, while the testing set is either (a) unchanged for ID testing, or (b) randomly rotated and rescaled for OOD testing. In order for all models to be trained on a single GPU, we choose a ResNet (He et al., 2016) with 16 layers as the baseline—this is different from previous works such as (Sosnovik et al., 2020) where a WideResNet baseline (Zagoruyko & Komodakis, 2016) is adopted—and we keep the number of trainable parameters almost the same for comparing networks. A group max-pooling is performed after the final residual block to achieve invariant representations.

Similar to the idea of (Sosnovik et al., 2020; Zhu et al., 2019), the data set is augmented (*without* rotation and rescaling) during training by randomly cropping a 12 pixel zero-padded image. Furthermore, random horizontal flipping and Cutout (DeVries & Taylor, 2017) with 32 pixels are applied to the cropped image. We train all models for 1000 epochs with a batch size of 64, using an SGD optimizer with Nesterov momentum set to 0.9 and weight decay set to $5 \times 10^{-4}$. Learning rate starts at 0.1, and is scheduled to decrease tenfold after 300, 400, 600, and 800 epochs.

The mean ± std after three independent trials of both ID and OOD test accuracy is displayed in Table 4. It can be observed that the proposed $\mathcal{RST}$-ResNet significantly outperforms all comparing models, especially for OOD generalization, demonstrating further its advantage in computer vision where both rotation and scale transformations are intrinsic symmetries of the learning task.

## 7 Conclusion

In this paper, we have proposed the roto-scale-translation equivariant CNN ($\mathcal{RST}$-CNN), which is able to achieve equivariance jointly over these three groups. Through truncated expansion of the joint convolutional filters under pre-fixed low-frequency spatial modes, which is motivated by a rigorous stability analysis of the representation, the proposed model provably attains deformation robust equivariance, i.e., the features stay "approximately" equivariant even if the $\mathcal{RST}$ transformation is "contaminated" by nuisance input distortion, a property that is crucial for out-of-distribution model generalization. Experiments on vision tasks with intrinsic $\mathcal{RST}$ symmetry are conducted to demonstrate the improved generalization capability of our proposed model under both in-distribution and out-of-distribution setting, especially in the small data regime.

One limitation of the current work is that we have considered deformation robust neural networks that are equivariant to only the *regular* representation of the group $\mathcal{RST}$. Such models have empirically proved to exhibit stronger generalization performance because of their ability to encode any function on the group. However, regular representation requires high dimensional feature spaces, and the memory consumption (mainly from storing intermediate features) of the proposed $\mathcal{RST}$-CNN is $N_r \times N_s$ times that of a regular CNN with the same number of unstructured channels. For future work, we will extend the idea in this

paper to construct deformation robust steerable $\mathcal{RST}$-CNNs with reduced model size and efficient network implementation.

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

## A  Appendix

## B  Proof of Proposition 1

We first recall for an input RGB image $x^{(0)}(u, \lambda)$, a roto-scale-translation transformation on this image can be understood as an $\mathcal{RST}$-action (representation) $D_g^{(0)} = D_{\eta,\beta,v}^{(0)}$ on $x^{(0)}$.

$$[D_{\eta,\beta,v}^{(0)} x^{(0)}](u, \lambda) = x^{(0)} \left( R_{-\eta} 2^{-\beta}(u - v), \lambda \right). \tag{18}$$

For the hidden layers $1 \leq l \leq L$, the action $D_{\eta,\beta,v}^{(l)}$ on the space $\mathcal{X}^{(l)}$ consisting of features in the form of $x^{(l)}(u, \theta, \alpha, \lambda)$ is defined as

$$[D_{\eta,\beta,v}^{(l)} x^{(l)}](u, \theta, \alpha, \lambda) = x^{(l)} \left( R_{-\eta} 2^{-\beta}(u - v), \theta - \eta, \alpha - \beta, \lambda \right), \; \forall l \geq 1. \tag{19}$$

We restate below Proposition 1 of the paper that outlines the general network architecture to achieve $\mathcal{RST}$-equivariance under the representations $D_{\eta,\beta,v}^{(l)}$, $0 \leq l \leq L$.

**Proposition 1.** *An $L$-layer feedforward neural network is $\mathcal{RST}$-equivariant under the representations* (18) (19) *if and only if the layer-wise operations are defined as* (20) *and* (21):

$$x^{(1)}[x^{(0)}](u, \theta, \alpha, \lambda) = \sigma \left[ \sum_{\lambda'} \int_{\mathbb{R}^2} x^{(0)}(u + u', \lambda') \cdot 2^{-2\alpha} W_{\lambda',\lambda}^{(1)} \left( 2^{-\alpha} R_{-\theta} u' \right) du' + b^{(1)}(\lambda) \right], \tag{20}$$

$$x^{(l)}[x^{(l-1)}](u, \theta, \alpha, \lambda) = \sigma \left[ \sum_{\lambda'} \int_{\mathbb{R}^2} \int_{S^1} \int_{\mathbb{R}} x^{(l-1)}(u + u', \theta + \theta', \alpha + \alpha', \lambda') \right.$$

$$\left. \cdot 2^{-2\alpha} W_{\lambda',\lambda}^{(l)} \left( 2^{-\alpha} R_{-\theta} u', \theta', \alpha' \right) d\alpha' d\theta' du' + b^{(l)}(\lambda) \right] \tag{21}$$

*where $\sigma : \mathbb{R} \to \mathbb{R}$ is a pointwise nonlinearity, $W_{\lambda',\lambda}^{(1)}(u)$ is the spatial convolutional filter in the first layer with output channel $\lambda$ and input channel $\lambda'$, $W_{\lambda',\lambda}^{(l)}(u, \theta, \alpha)$ is the $\mathcal{RST}$ joint convolutional filter for layer $l > 1$, and $\int_{S^1} f(\alpha) d\alpha$ denotes the normalized $S^1$ integral $\frac{1}{2\pi} \int_0^{2\pi} f(\alpha) d\alpha$.*

### B.1 Sufficient Part of the Proof of Proposition 1

*Proof.* We first prove the sufficient part of Proposition 1. That is, given the layer-wise definition of the $L$-layer feedforward neural network (20) and (21), $\mathcal{RST}$-equivariance under the regular representation (18) (19) can be achieved, i.e.,

$$x^{(l)}[D^{(l-1)}_{\eta,\beta,v}x^{(l-1)}] = D^{(l)}_{\eta,\beta,v}x^{(l)}[x^{(l-1)}], \quad \forall l \geq 1. \tag{22}$$

Indeed, for the first layer, i.e., $l = 1$, we write the LHS of (22) as

$$x^{(1)}[D^{(0)}_{\eta,\beta,v}x^{(0)}](u,\theta,\alpha,\lambda) = \sigma\left(\sum_{\lambda'}\int_{\mathbb{R}^2} 2^{-2\alpha}D^{(0)}_{\eta,\beta,v}x^{(0)}(u+u',\lambda')W^{(1)}_{\lambda',\lambda}(2^{-\alpha}R_{-\theta}u')du' + b^{(1)}(\lambda)\right)$$

$$= \sigma\left(\sum_{\lambda'}\int_{\mathbb{R}^2} 2^{-2\alpha}x^{(0)}\left(R_{-\eta}2^{-\beta}(u+u'-v),\lambda'\right)W^{(1)}_{\lambda',\lambda}(2^{-\alpha}R_{-\theta}u')du' + b^{(1)}(\lambda)\right), \tag{23}$$

The RHS of (22) when $l = 1$ is

$$D^{(1)}_{\eta,\beta,v}x^{(1)}[x^{(0)}](u,\theta,\alpha,\lambda) = x^{(1)}[x^{(0)}]\left(R_{-\eta}2^{-\beta}(u-v),\theta-\eta,\alpha-\beta,\lambda\right)$$

$$= \sigma\left(\sum_{\lambda'}\int_{\mathbb{R}^2} 2^{-2(\alpha-\beta)}x^{(0)}\left(R_{-\eta}2^{-\beta}(u-v)+\tilde{u},\lambda'\right)W^{(1)}_{\lambda',\lambda}(2^{-(\alpha-\beta)}R_{\eta-\theta}\tilde{u})d\tilde{u} + b^{(1)}(\lambda)\right)$$

$$= \sigma\left(\sum_{\lambda'}\int_{\mathbb{R}^2} 2^{-2\alpha}2^{2\beta}x^{(0)}\left(R_{-\eta}2^{-\beta}(u-v)+2^{-\beta}R_{-\eta}u',\lambda'\right)W^{(1)}_{\lambda',\lambda}(2^{-\alpha}R_{-\theta}u')2^{-2\beta}du' + b^{(1)}(\lambda)\right) \tag{24}$$

$$= \sigma\left(\sum_{\lambda'}\int_{\mathbb{R}^2} 2^{-2\alpha}x^{(0)}\left(R_{-\eta}2^{-\beta}(u+u'-v),\lambda'\right)W^{(1)}_{\lambda',\lambda}(2^{-\alpha}R_{-\theta}u')du' + b^{(1)}(\lambda)\right), \tag{25}$$

where (24) uses the change of variable $u' = 2^{-\beta}R_{-\eta}\tilde{u}$. Thus (23) and (25) implies that (22) holds valid for $l = 1$.

For $l > 1$, with the definition of the $l$-th layer operation (21), the LHS of (22) becomes

$$x^{(l)}[D^{(l-1)}_{\eta,\beta,v}x^{(l-1)}](u,\theta,\alpha,\lambda)$$

$$= \sigma\left(\sum_{\lambda'}\int\int\int 2^{-2\alpha}D^{(l-1)}_{\eta,\beta,v}x^{(l-1)}\left(u+u',\theta+\theta',\alpha+\alpha',\lambda'\right)W^{(l)}_{\lambda',\lambda}\left(2^{-\alpha}R_{-\theta}u',\theta',\alpha'\right)d\alpha'd\theta'du' + b^{(l)}(\lambda)\right)$$

$$\tag{26}$$

$$= \sigma\left(\sum_{\lambda'}\int\int\int 2^{-2\alpha}x^{(l-1)}\left(R_{-\eta}2^{-\beta}(u+u'-v),\theta+\theta'-\eta,\alpha+\alpha'-\beta,\lambda'\right)\right.$$

$$\left. \cdot W^{(l)}_{\lambda',\lambda}\left(2^{-\alpha}R_{-\theta}u',\theta',\alpha'\right)d\alpha'd\theta'du' + b^{(l)}(\lambda)\right). \tag{27}$$

On the other hand, the RHS of (22) is

$$
D^{(l)}_{\eta,\beta,v} x^{(l)}[x^{(l-1)}](u,\theta,\alpha,\lambda)
$$
$$
= x^{(l)}[x^{(l-1)}]\left(R_{-\eta}2^{-\beta}(u-v),\theta-\eta,\alpha-\beta,\lambda\right) \tag{28}
$$
$$
= \sigma\Bigg(\sum_{\lambda'}\int\int\int 2^{-2(\alpha-\beta)}x^{(l-1)}\left(R_{-\eta}2^{-\beta}(u-v)+\tilde{u},\theta-\eta+\theta',\alpha-\beta+\alpha',\lambda'\right)
$$
$$
\cdot W^{(l)}_{\lambda'\lambda}\left(2^{-(\alpha-\beta)}R_{\eta-\theta}\tilde{u},\theta',\alpha'\right)d\alpha'd\theta'd\tilde{u}+b^{(l)}(\lambda)\Bigg) \tag{29}
$$
$$
= \sigma\Bigg(\sum_{\lambda'}\int\int\int 2^{-2\alpha}2^{2\beta}x^{(l-1)}\left(R_{-\eta}2^{-\beta}(u-v)+R_{-\eta}2^{-\beta}u',\theta-\eta+\theta',\alpha-\beta+\alpha',\lambda'\right)
$$
$$
\cdot W^{(l)}_{\lambda',\lambda}\left(2^{-\alpha}R_{-\theta}u',\theta',\alpha'\right)2^{-2\beta}d\alpha'd\theta'd\tilde{u}+b^{(l)}(\lambda)\Bigg), \tag{30}
$$

where the last equation again uses the same change of variable. Equation (27) combined with (30) implies that (22) holds true for all $l>1$. □

## B.2   Necessary Part of the Proof of Proposition 1

*Proof.* We first note that a general feedforward neural network propagating through the feature spaces $\mathcal{X}^{(l)}$ has the following form: when $l=1$,

$$
x^{(1)}[x^{(0)}](u,\theta,\alpha,\lambda)=\sigma\Bigg(\sum_{\lambda'}\int_{\mathbb{R}^2}x^{(0)}(u+u')W^{(1)}(u',\lambda',u,\theta,\alpha,\lambda)du'+b^{(1)}(\lambda)\Bigg), \tag{31}
$$

and, for $l>1$,

$$
x^{(l)}[x^{(l-1)}](u,\theta,\alpha,\lambda)=\sigma\Bigg(\sum_{\lambda'}\int_{\mathbb{R}^2}\int_{S^1}\int_{\mathbb{R}}x^{(l-1)}(u+u',\theta+\theta',\alpha+\alpha',\lambda')
$$
$$
\cdot W^{(l)}(u',\theta',\alpha',\lambda',u,\theta,\alpha,\lambda)d\alpha'd\theta'du'+b^{(l)}(\lambda)\Bigg). \tag{32}
$$

To prove the necessary part of Proposition 1, we want to verify that $\mathcal{RST}$-equivariance (22) implies that the weight matrices $W^{(l)}$ in (31) and (32) take the special convolutional form in (20) and (21).

Indeed, when $l=1$, the RHS of (22) under the operation (31) is

$$
D^{(1)}_{\eta,\beta,v} x^{(1)}[x^{(0)}](u,\theta,\alpha,\lambda) \tag{33}
$$
$$
= x^{(1)}[x^{(0)}]\left(R_{-\eta}2^{-\beta}(u-v),\theta-\eta,\alpha-\beta,\lambda\right) \tag{34}
$$
$$
= \sigma\Bigg(\sum_{\lambda'}\int x^{(0)}(R_{-\eta}2^{-\beta}(u-v)+\tilde{u},\lambda')W^{(1)}(\tilde{u},\lambda',R_{-\eta}2^{-\beta}(u-v),\theta-\eta,\alpha-\beta,\lambda)d\tilde{u}+b^{(1)}(\lambda)\Bigg) \tag{35}
$$
$$
= \sigma\Bigg(\sum_{\lambda'}\int 2^{-2\beta}x^{(0)}(R_{-\eta}2^{-\beta}(u+u'-v),\lambda')
$$
$$
\cdot W^{(1)}\left(R_{-\eta}2^{-\beta}u',\lambda',R_{-\eta}2^{-\beta}(u-v),\theta-\eta,\alpha-\beta,\lambda\right)du'+b^{(1)}(\lambda)\Bigg), \tag{36}
$$

with change of variable similar to the sufficient part. For the RHS, we have

$$x^{(1)}[D_{\eta,\beta,v}^{(0)}x^{(0)}](u,\theta,\alpha,\lambda) \tag{37}$$

$$=\sigma\left(\sum_{\lambda'}\int D_{\eta,\beta,v}^{(1)}x^{(0)}(u+u',\lambda')W^{(1)}(u',\lambda',u,\theta,\alpha,\lambda)du'+b^{(1)}(\lambda)\right) \tag{38}$$

$$=\sigma\left(\sum_{\lambda'}\int x^{(0)}(R_{-\eta}2^{-\beta}(u+u'-v),\lambda')W^{(1)}(u',\lambda',u,\theta,\alpha,\lambda)du'+b^{(1)}(\lambda)\right) \tag{39}$$

Hence, for (22) to hold when $l=1$, we need

$$W^{(1)}(u',\lambda',u,\theta,\alpha,\lambda) = 2^{-2\beta}W^{(1)}(R_{-\eta}2^{-\beta}u',\lambda',R_{-\eta}2^{-\beta}(u-v),\theta-\eta,\alpha-\beta,\lambda), \tag{40}$$

For all $u,\theta,\alpha,\lambda,u',\lambda',v,\eta,\beta$. Keeping $u,\theta,\alpha,\lambda,u',\lambda',\eta,\beta$ fixed while varying $v$ in (40), we deduce that $W^{(1)}(u',\lambda',u,\theta,\alpha,\lambda)$ does not depend on the third variable $u$. Hence $W^{(1)}(u',\lambda',u,\theta,\alpha,\lambda) = W^{(1)}(u',\lambda',0,\theta,\alpha,\lambda),\forall u\in\mathbb{R}^2$. Further define $W_{\lambda,\lambda'}^{(1)}(u')$ as

$$W_{\lambda,\lambda'}^{(1)}(u') = W^{(1)}(u',\lambda',0,0,0,\lambda). \tag{41}$$

Therefore, for any fixed $u',\lambda',u,\theta,\alpha,\lambda$, setting $\beta=\alpha, \eta=\theta$ in (40) yields

$$W^{(1)}(u',\lambda',u,\theta,\alpha,\lambda) = W^{(1)}(R_{-\theta}2^{-\alpha}u',\lambda',R_{-\theta}2^{-\alpha}(u-v),\theta-\theta,\alpha-\alpha,\lambda)2^{-2\alpha} \tag{42}$$

$$= W^{(1)}(R_{-\theta}2^{-\alpha}u',\lambda',0,0,0,\lambda)2^{-2\alpha} = W_{\lambda,\lambda'}^{(1)}(R_{-\theta}2^{-\alpha}u')2^{-2\alpha} \tag{43}$$

Hence (31) has the special form (20).

For the subsequent layers $l>1$, a similar argument yields

$$W^{(l)}(u',\theta',\alpha',\lambda',u,\theta,\alpha,\lambda) = W^{(l)}(R_{-\eta}2^{-\beta}u',\theta',\alpha',\lambda',R_{-\eta}2^{-\beta}(u-v),\theta-\eta,\alpha-\beta,\lambda)2^{-2\beta}, \tag{44}$$

for all $u,\theta,\alpha,\lambda,u',\theta',\alpha',\lambda',v,\eta,\beta$. Similarly, we can keep $u,\theta,\alpha,\lambda,u',\theta',\alpha',\lambda',\eta,\beta$ fixed while varying $v$ in (44), which implies that $W^{(l)}(u',\theta',\alpha',\lambda',u,\theta,\alpha,\lambda)$ does not depend on the fifth variable $u$. Again, let us define

$$W_{\lambda,\lambda'}^{(l)}(u',\theta',\alpha') = W^{(l)}(u',\theta',\alpha',\lambda',0,0,0,\lambda). \tag{45}$$

For any given $u',\theta',\alpha',\lambda',u,\theta,\alpha,\lambda$, setting $\beta=\alpha, \eta=\theta$ leads us to

$$W^{(l)}(u',\theta',\alpha',\lambda',u,\theta,\alpha,\lambda) = W^{(l)}(R_{-\theta}2^{-\alpha}u',\theta',\alpha',\lambda',R_{-\theta}2^{-\alpha}(u-v),0,0,\lambda)2^{-2\alpha}$$

$$=W^{(l)}(R_{-\theta}2^{-\alpha}u',\theta',\alpha',\lambda',0,0,0,\lambda)2^{-2\alpha} = W_{\lambda',\lambda}^{(l)}(2^{-\alpha}u',\theta'-\theta,\alpha'-\alpha), \tag{46}$$

which implies that (32) can be written in the form of (21). This concludes the proof of Proposition 1. □

## C   Proof of Theorem 1

We prove, in this section, the deformation stability of the $\mathcal{RST}$-CNN (Theorem 1 of the paper) under the following three assumptions:

**(A1):** The pointwise nonlinearity $\sigma:\mathbb{R}\to\mathbb{R}$ is non-expansive, i.e., $|\sigma(x)-\sigma(y)|\leq|x-y|, \ \forall x,y\in\mathbb{R}$.

**(A2):** The convolutional filters are bounded in the following sense: $A_l\leq 1, \forall l\geq 1$, where

$$A_1 := \pi\max\left\{\sup_\lambda\sum_{\lambda'}\|a_{\lambda',\lambda}^{(1)}\|_{\mathrm{FB}}, \frac{M_0}{M_1}\sup_{\lambda'}\sum_{\lambda'}\|a_{\lambda',\lambda}^{(1)}\|_{\mathrm{FB}}\right\} \tag{47}$$

$$A_l := \pi\max\left\{\sup_\lambda\sum_{\lambda'}\sum_n\|a_{\lambda',\lambda}^{(l)}(\cdot,n)\|_{\mathrm{FB}}, \frac{2M_{l-1}}{M_l}\sum_n\sup_{\lambda'}\sum_\lambda\|a_{\lambda',\lambda}^{(l)}(\cdot,n)\|_{\mathrm{FB}}\right\}, \ \forall l>1, \tag{48}$$

in which the FB-norm $\| \cdot \|_{\mathrm{FB}}$ of a sequence $\{a(k)\}_{k \geq 0}$ and double sequence $\{b(k,m)\}_{k,m \geq 0}$ is the weighted $l_2$-norm defined as

$$\|a\|_{\mathrm{FB}}^2 = \sum_k \mu_k a(k)^2, \quad \|b\|_{\mathrm{FB}}^2 = \sum_k \sum_m \mu_k b(k,m)^2, \tag{49}$$

with $\mu_k$ being the eigenvalues of the Dirichlet Laplacian on a unit disk.

**(A3):** The input distortion is small. More specifically, let

$$D_\tau x^{(0)}(u,\lambda) = x^{(0)}(\rho(u),\lambda), \quad \text{and} \quad D_\tau x^{(l)}(u,\theta,\alpha,\lambda) = x^{(l)}(\rho(u),\theta,\alpha,\lambda), \quad l \geq 1, \tag{50}$$

where $\rho(u) = u - \tau(u)$, and $\tau : \mathbb{R}^2 \to \mathbb{R}^2$ is a $C^2$ local (spatial) distortion. We assume

$$|\nabla \tau|_\infty := \sup_u \|\nabla \tau(u)\| < 1/5, \tag{51}$$

with $\| \cdot \|$ being the operator norm.

We repeat below Theorem 1 of the paper that quantifies the deformation stability of the equivariant representation in an $\mathcal{RST}$-CNN under the assumptions **(A1)-(A3)**:

**Theorem 2.** *Let $D_{\eta,\beta,v}^{(l)}$, $0 \leq l \leq L$, be the $\mathcal{RST}$ group actions defined in (18)(19), and let $D_\tau$ be a small input deformation define in (50). If an $\mathcal{RST}$-CNN satisfies the assumptions **(A1)-(A3)**, we have, for any $L$,*

$$\left\| x^{(L)}[D_{\eta,\beta,v}^{(0)} \circ D_\tau x^{(0)}] - D_{\eta,\beta,v}^{(L)} x^{(L)}[x^{(0)}] \right\| \leq 2^{\beta+1} \left( 4L|\nabla \tau|_\infty + 2^{-j_L}|\tau|_\infty \right) \|x^{(0)}\|, \tag{52}$$

*where the layer-wise feature norm is defined as as*

$$\begin{cases} \left\| x^{(0)} \right\|^2 = \dfrac{1}{M_0} \sum_{\lambda=1}^{M_0} \int_{\mathbb{R}^2} |x^{(0)}(u,\lambda)|^2 du, \\ \left\| x^{(l)} \right\|^2 = \sup_\alpha \dfrac{1}{M_l} \sum_{\lambda=1}^{M_l} \iint_{S^1 \times \mathbb{R}^2} |x^{(l)}(u,\theta,\alpha,\lambda)|^2 du d\theta, \quad \forall l \geq 1 \end{cases} \tag{53}$$

The proof of Theorem 1 follows the similar steps in (Zhu et al., 2019). More specifically, we aim to establish the following two propositions that show the layer-wise non-expansiveness of the model (Proposition 2) and quantify the perturbation of equivariance with the presence of layer-wise spatial deformation $D_\tau$ (Proposition 3).

**Proposition 2.** *Under **(A1)** and **(A2)**, an $\mathcal{RST}$-CNN satisfies:*

(a) *For any $l \geq 1$, the $l$-th layer mapping $x^{(l)}[\cdot]$ defined in (21) is non-expansive, i.e.,*

$$\|x^{(l)}[x_1] - x^{(l)}[x_2]\| \leq \|x_1 - x_2\|, \quad \forall x_1, x_2. \tag{54}$$

(b) *Let $x_0^{(l)}(u,\theta,\alpha,\lambda)$ be the $l$-th layer output given a zero input $x^{(0)}(u,\lambda) = 0$, then $x_0^{(l)}(u,\theta,\alpha,\lambda)$ depends only on $\lambda$, i.e., $x_0^{(l)}(u,\theta,\alpha,\lambda) = x_0^{(l)}(\lambda)$.*

(c) *Let $x_c^{(l)}$ be the centered version of $x^{(l)}$ after subtracting $x_0^{(l)}$, i.e.,*

$$x_c^{(0)}(u,\lambda) := x^{(0)}(u,\lambda) - x_0^{(0)}(\lambda) = x^{(0)}(u,\lambda), \ x_c^{(l)}(u,\theta,\alpha,\lambda) := x^{(l)}(u,\theta,\alpha,\lambda) - x_0^{(l)}(\lambda), \ l \geq 1, \tag{55}$$

*then $\|x_c^{(l)}\| \leq \|x_c^{(l-1)}\|$, $\forall l \geq 1$. As a result, $\|x_c^{(l)}\| \leq \|x_c^{(0)}\| = \|x^{(0)}\|$.*

**Proposition 3.** *In an $\mathcal{RST}$-CNN satisfying **(A1)** to **(A3)**, the following statements hold true.*

(a) Given any $l \geq 1$,

$$\left\| x^{(l)}[D_\tau x^{(l-1)}] - D_\tau x^{(l)}[x^{(l-1)}] \right\| \leq 8|\nabla\tau|_\infty \left\| x_c^{(l-1)} \right\|, \tag{56}$$

where $x_c^{(l-1)}$ is defined in (55)

(b) Given any $l \geq 1$, we have

$$\left\| D_{\eta,\beta,v}^{(l)} x^{(l)} \right\| = 2^\beta \left\| x^{(l)} \right\|, \tag{57}$$

and

$$\left\| x^{(l)}[D_{\eta,\beta,v}^{(l-1)} \circ D_\tau x^{(l-1)}] - D_{\eta,\beta,v}^{(l)} D_\tau x^{(l)}[x^{(l-1}] \right\| \leq 2^{\beta+3}|\nabla\tau|_\infty \left\| x_c^{(l-1)} \right\|. \tag{58}$$

(c) For any $l \geq 1$,

$$\left\| x^{(l)}[D_{\eta,\beta,v}^{(0)} \circ D_\tau x^{(l-1)}] - D_{\eta,\beta,v}^{(l)} D_\tau x^{(l)}[x^{(0)}] \right\| \leq 2^{\beta+3} l|\nabla\tau|_\infty \left\| x_c^{(0)} \right\|. \tag{59}$$

(d) For any $l \geq 1$,

$$\left\| D_{\eta,\beta,v}^{(l)} D_\tau x^{(l)} - D_{\eta,\beta,v}^{(l)} x^{(l)} \right\| \leq 2^{\beta+1-j_l}|\tau|_\infty \left\| x_c^{(l-1)} \right\| \leq 2^{\beta+1-j_l}|\tau|_\infty \left\| x^{(0)} \right\|. \tag{60}$$

## C.1 Proof of Proposition 2

Before proving Proposition 2, we present the following two lemmas that are crucial to bound various norms of the convolutional filters using their Fourier-Bessle (FB) norm

**Lemma 1.** *Let $\{\psi_k\}_k$ be the Fourier-Bessel basis on the unit disk $D \subset \mathbb{R}^2$, and let $\{\varphi_m\}_m$ be the Fourier basis on the unit circle $S^1$. Assume that*

$$F(u) = \sum_k a(k)\psi_k(u), \quad G(u,\theta) = \sum_k \sum_m b(k,m)\psi_k(u)\varphi_m(\theta) \tag{61}$$

*are functions in $H_0^1(2^j D)$ and $H_0^1(2^j D) \times L^2(S^1)$, respectively. Then*

$$\int |F(u)|du, \quad \int |u||\nabla F(u)|du, \quad \int |\nabla F(u)|du \leq \pi\|a\|_{FB} = \pi \left( \sum_k \mu_k a(k)^2 \right)^{1/2}, \tag{62}$$

$$\iint |G(u,\theta)|dud\theta, \quad \iint |u||\nabla_u G(u,\theta)|dud\theta, \quad \iint |\nabla_u G(u)|dud\theta \leq \pi\|b\|_{FB} = \pi \left( \sum_{k,m} \mu_k b(k,m)^2 \right)^{1/2}. \tag{63}$$

The proof of Lemma 1 can be found in Proposition A.1 of (Cheng et al., 2019). A direct application of Lemma 1 leads to the following lemma.

**Lemma 2.** *Let $a_{\lambda',\lambda}^{(l)}(k,m,n)$ be the coefficients of the filter $W_{\lambda',\lambda}^{(l)}(u,\theta,\alpha)$ (supported on $2^{j_l} D \times S^1 \times [-1,1]$) under the separable bases $\{\psi_{j_l,k}(u)\}_k$, $\{\varphi_m(\theta)\}_m$ and $\{\xi_n(\alpha)\}_n$ defined in the main paper, and define $W_{\lambda',\lambda,n}^{(l)}(u,\theta)$ as*

$$W_{\lambda',\lambda,n}^{(l)}(u,\theta) := \sum_k \sum_m a_{\lambda',\lambda}^{(l)}(k,m,n)\psi_{j_l,k}(u)\varphi_m(\theta). \tag{64}$$

*We have*

$$B_{\lambda',\lambda}^{(1)}, C_{\lambda',\lambda}^{(1)}, 2^{j_1} D_{\lambda',\lambda}^{(1)} \leq \pi\|a_{\lambda',\lambda}^{(1)}\|_{FB}, \quad B_{\lambda',\lambda,n}^{(l)}, C_{\lambda',\lambda,n}^{(l)}, 2^{j_l} D_{\lambda',\lambda,n}^{(l)} \leq \pi\|a_{\lambda',\lambda}^{(l)}(\cdot,n)\|_{FB}, \; \forall l > 1, \tag{65}$$

*where*

$$
\begin{cases}
B_{\lambda',\lambda}^{(1)} := \int \left| W_{\lambda',\lambda}^{(1)}(u) \right| du, & B_{\lambda',\lambda,n}^{(l)} := \int_{S^1} \int_{\mathbb{R}^2} \left| W_{\lambda',\lambda,n}^{(l)}(u,\theta) \right| du d\theta, \quad l > 1, \\[2mm]
C_{\lambda',\lambda}^{(1)} := \int |u| \left| \nabla_u W_{\lambda',\lambda}^{(1)}(u) \right| du, & C_{\lambda',\lambda,n}^{(l)} := \int_{S^1} \int_{\mathbb{R}^2} |u| \left| \nabla_u W_{\lambda',\lambda,n}^{(l)}(u,\theta) \right| du d\theta, \quad l > 1, \\[2mm]
D_{\lambda',\lambda}^{(1)} := \int \left| \nabla_u W_{\lambda',\lambda}^{(1)}(u) \right| du, & D_{\lambda',\lambda,n}^{(l)} := \int_{S^1} \int_{\mathbb{R}^2} \left| \nabla_u W_{\lambda',\lambda,n}^{(l)}(u,\theta) \right| du d\theta, \quad l > 1.
\end{cases}
\tag{66}
$$

*Hence we have*

$$
B_l, C_l, 2^{j_l} D_l \le A_l,
\tag{67}
$$

*where*

$$
\begin{aligned}
B_1 &:= \max \left\{ \sup_\lambda \sum_{\lambda'=1}^{M_0} B_{\lambda',\lambda}^{(1)}, \ \frac{M_0}{M_1} \sup_{\lambda'} \sum_{\lambda=1}^{M_1} B_{\lambda',\lambda}^{(1)} \right\}, \\
C_1 &:= \max \left\{ \sup_\lambda \sum_{\lambda'=1}^{M_0} C_{\lambda',\lambda}^{(1)}, \ \frac{M_0}{M_1} \sup_{\lambda'} \sum_{\lambda=1}^{M_1} C_{\lambda',\lambda}^{(1)} \right\}, \\
D_1 &:= \max \left\{ \sup_\lambda \sum_{\lambda'=1}^{M_0} D_{\lambda',\lambda}^{(1)}, \ \frac{M_0}{M_1} \sup_{\lambda'} \sum_{\lambda=1}^{M_1} D_{\lambda',\lambda}^{(1)} \right\},
\end{aligned}
\tag{68}
$$

*and, for $l > 1$,*

$$
\begin{aligned}
B_l &:= \max \left\{ \sup_\lambda \sum_{\lambda'=1}^{M_{l-1}} \sum_n B_{\lambda',\lambda,n}^{(l)}, \ \frac{2M_{l-1}}{M_l} \sum_n B_{l,n} \right\}, \quad B_{l,n} := \sup_{\lambda'} \sum_{\lambda=1}^{M_l} B_{\lambda',\lambda,n}^{(l)}, \\
C_l &:= \max \left\{ \sup_\lambda \sum_{\lambda'=1}^{M_{l-1}} \sum_n C_{\lambda',\lambda,n}^{(l)}, \ \frac{2M_{l-1}}{M_l} \sum_n C_{l,n} \right\}, \quad C_{l,n} := \sup_{\lambda'} \sum_{\lambda=1}^{M_l} C_{\lambda',\lambda,n}^{(l)}, \\
D_l &:= \max \left\{ \sup_\lambda \sum_{\lambda'=1}^{M_{l-1}} \sum_n D_{\lambda',\lambda,n}^{(l)}, \ \frac{2M_{l-1}}{M_l} \sum_n D_{l,n} \right\}, \quad D_{l,n} := \sup_{\lambda'} \sum_{\lambda=1}^{M_l} D_{\lambda',\lambda,n}^{(l)}.
\end{aligned}
\tag{69}
$$

*The bound on $A_l, \forall l \ge 1$, i.e., (A2), therefore implies that*

$$
B_l, C_l, 2^{j_l} D_l \le 1, \quad \forall l \ge 1.
\tag{70}
$$

*Proof of Lemma 2.* Applying Lemma 1 to the filters $W_{\lambda',\lambda}^{(1)}(u)$ and $W_{\lambda',\lambda,n}^{(l)}(u,\theta)$ defined in (64) after rescaling the spatial variable $u$ easily leads to the desired bounds (67). The rest of the lemma follows from the assumption **(A2)**. $\qquad\square$

*Proof of Proposition 2.* To avoid cumbersome notations, we drop the layer index $(l)$ in the filters $W_{\lambda',\lambda}^{(l)}$ and $b^{(l)}$, and let $M = M_l, M' = M_{l-1}$ when the context is clear. The proof of (a) is similar to that of Proposition 2(a) in (Zhu et al., 2019) after a further integration on $S^1$. More specifically, when $l = 1$, the definition of $B_1$ in (66) implies that

$$
\sup_\lambda \sum_{\lambda'} B_{\lambda',\lambda}^{(1)} \le B_1, \quad \sup_{\lambda'} \sum_\lambda B_{\lambda',\lambda}^{(1)} \le B_1 \frac{M}{M'}
\tag{71}
$$

Therefore, given two inputs $x_1$ and $x_2$, we have

$$\left| \left( x^{(1)}[x_1] - x^{(1)}[x_2] \right) (u, \theta, \alpha, \lambda) \right|^2 \tag{72}$$

$$= \left| \sigma \left( \sum_{\lambda'} \int x_1(u + u', \lambda') W_{\lambda', \lambda} \left( 2^{-\alpha} R_{-\theta} u' \right) 2^{-2\alpha} du' + b(\lambda) \right) \right.$$

$$\left. - \sigma \left( \sum_{\lambda'} \int x_2(u + u', \lambda') W_{\lambda', \lambda} \left( 2^{-\alpha} R_{-\theta} u' \right) 2^{-2\alpha} du' + b(\lambda) \right) \right|^2 \tag{73}$$

$$\leq \left| \sum_{\lambda'} \int x_1(u + u', \lambda') W_{\lambda', \lambda} \left( 2^{-\alpha} R_{-\theta} u' \right) 2^{-2\alpha} du' - \sum_{\lambda'} \int x_2(u + u', \lambda') W_{\lambda', \lambda} \left( 2^{-\alpha} R_{-\theta} u' \right) 2^{-2\alpha} du' \right|^2 \tag{74}$$

$$= \left| \sum_{\lambda'} \int (x_1 - x_2)(u + u', \lambda') W_{\lambda', \lambda} \left( 2^{-\alpha} R_{-\theta} u' \right) 2^{-2\alpha} du' \right|^2 \tag{75}$$

$$\leq \left( \sum_{\lambda'} \int |(x_1 - x_2)(u + u', \lambda')|^2 \left| W_{\lambda', \lambda}(2^{-\alpha} R_{-\theta} u') \right| 2^{-2\alpha} du' \right) \sum_{\lambda'} \int \left| W_{\lambda', \lambda}(2^{-\alpha} R_{-\theta} u') \right| 2^{-2\alpha} du' \tag{76}$$

$$= \left( \sum_{\lambda'} \int |(x_1 - x_2)(u + u', \lambda')|^2 \left| W_{\lambda', \lambda}(2^{-\alpha} R_{-\theta} u') \right| 2^{-2\alpha} du' \right) \left( \sum_{\lambda'} B_{\lambda', \lambda}^{(1)} \right) \tag{77}$$

$$\leq B_1 \sum_{\lambda'} \int |(x_1 - x_2)(\tilde{u}, \lambda')|^2 \left| W_{\lambda', \lambda}(2^{-\alpha} R_{-\theta} (\tilde{u} - u)) \right| 2^{-2\alpha} d\tilde{u} \tag{78}$$

Hence, given any $\alpha \in \mathbb{R}$, we have

$$\sum_{\lambda} \int_{S^1} \int_{\mathbb{R}^2} \left| \left( x^{(1)}[x_1] - x^{(1)}[x_2] \right) (u, \alpha, \theta, \lambda) \right|^2 du d\theta$$

$$\leq \sum_{\lambda} \int_{S^1} \int_{\mathbb{R}^2} B_1 \sum_{\lambda'} \int |(x_1 - x_2)(\tilde{u}, \lambda')|^2 \left| W_{\lambda', \lambda}(2^{-\alpha} R_{-\theta} (\tilde{u} - u)) \right| 2^{-2\alpha} d\tilde{u} du d\theta \tag{79}$$

$$= B_1 \sum_{\lambda'} \int_{\mathbb{R}^2} |(x_1 - x_2)(\tilde{u}, \lambda')|^2 \left( \sum_{\lambda} \int_{S^1} \int_{\mathbb{R}^2} \left| W_{\lambda', \lambda}(2^{-\alpha} R_{-\theta} (\tilde{u} - u)) \right| 2^{-2\alpha} du d\theta \right) d\tilde{u} \tag{80}$$

$$= B_1 \sum_{\lambda'} \int |(x_1 - x_2)(\tilde{u}, \lambda')|^2 \left( \sum_{\lambda} B_{\lambda', \lambda}^{(1)} \right) d\tilde{u} \tag{81}$$

$$\leq B_1^2 \frac{M}{M'} \sum_{\lambda'} \int |(x_1 - x_2)(\tilde{u}, \lambda')|^2 d\tilde{u} \tag{82}$$

$$= B_1^2 M \|x_1 - x_2\|^2 \tag{83}$$

$$\leq M \|x_1 - x_2\|^2, \tag{84}$$

where the last inequality comes from Lemma 2, and (80) makes use of the fact that

$$\int_{\mathbb{R}^2} \left| W(2^{-\alpha} R_{-\theta} u) \right| 2^{-2\alpha} du = \int_{\mathbb{R}^2} |W(u)| du, \quad \forall \alpha \in \mathbb{R}, \ \forall \theta \in S^1, \tag{85}$$

and $\int_{S^1} d\theta = 1$ due to the normalization factor $1/2\pi$ in the definition. Therefore, we have

$$\|x^{(1)}[x_1] - x^{(1)}[x_2]\|^2 = \sup_{\alpha} \frac{1}{M} \sum_{\lambda} \iint \left| \left( x^{(1)}[x_1] - x^{(1)}[x_2] \right) (u, \theta, \alpha, \lambda) \right|^2 du d\theta \leq \|x_1 - x_2\|^2. \tag{86}$$

This concludes the proof of (a) for the case $l = 1$. For the case $l > 1$, the same technique applies by considering the joint convolution $\int_{S^1} \int_{\mathbb{R}^2} (\cdots) du d\theta$ while making use of (85), and we omit the detail.

For part (b), we use mathematical induction. More specifically, $x_0^{(0)}(u, \lambda) = 0$ by definition. For $l = 1$, $x_0^{(1)}(u, \theta, \alpha, \lambda) = \sigma(b^{(1)}(\lambda))$. Assuming that $x_0^{(l-1)}(u, \theta, \alpha, \lambda) = x_0^{(l-1)}(\lambda)$ for some $l > 1$, we have

$$x_0^{(l)}(u, \alpha, \lambda)$$

$$= \sigma \left( \sum_{\lambda'} \int_{S^1} \int_{\mathbb{R}^2} \int_{\mathbb{R}} x_0^{(l-1)}(u + u', \theta + \theta', \alpha + \alpha', \lambda') W_{\lambda', \lambda}^{(l)} \left( 2^{-\alpha} R_{-\theta} u', \theta', \alpha' \right) 2^{-2\alpha} d\alpha' du' d\theta' + b^{(l)}(\lambda) \right) \quad (87)$$

$$= \sigma \left( \sum_{\lambda'} x_0^{(l-1)}(\lambda') \int_{S^1} \int_{\mathbb{R}^2} \int_{\mathbb{R}} W_{\lambda', \lambda}^{(l)} \left( 2^{-\alpha} R_{-\theta} u', \theta', \alpha' \right) 2^{-2\alpha} d\alpha' du' d\theta' + b^{(l)}(\lambda) \right) \quad (88)$$

$$= \sigma \left( \sum_{\lambda'} x_0^{(l-1)}(\lambda') \int_{S^1} \int_{\mathbb{R}^2} \int_{\mathbb{R}} W_{\lambda', \lambda}^{(l)} \left( u', \theta', \alpha' \right) d\alpha' du' d\theta' + b^{(l)}(\lambda) \right) \quad (89)$$

$$= x_0^{(l)}(\lambda). \quad (90)$$

To prove part (c): for any $l > 1$, we have

$$\|x_c^{(l)}\| = \|x^{(l)} - x_0^{(l)}\| = \|x^{(l)}[x^{(l-1)}] - x_0^{(l)}[x_0^{(l-1)}]\| \leq \|x^{(l-1)} - x_0^{(l-1)}\| = \|x_c^{(l-1)}\|, \quad (91)$$

where the inequality comes from the layer-wise non-expansiveness (54) in part (a). An easy induction leads to (c). $\quad \square$

## C.2    Proof of Proposition 3

*Proof.* Just like Proposition 2(a), the proofs of part (a) and part (d), respectively, of Proposition 3 are similar to those of Proposition 3(a) and Proposition 4 of (Zhu et al., 2019). More specifically, making use of (85) and $\int_{S^1} d\theta = 1$ when $l = 1$, and further taking the integral $\int_{S^1} \int_{\mathbb{R}^2} (\cdots) du d\theta$ over $S^1 \times \mathbb{R}^2$ instead of just $\mathbb{R}^2$ when $l > 1$, we arrive at the following

$$\left\| x^{(l)}[D_\tau x^{(l-1)}] - D_\tau x^{(l)}[x^{(l-1)}] \right\| \leq 4(B_l + C_l)|\nabla\tau|_\infty \left\| x_c^{(l-1)} \right\| \leq 4(B_l + C_l)|\nabla\tau|_\infty \left\| x^{(0)} \right\|, \quad \forall l \geq 1, \quad (92)$$

$$\left\| D_{\eta, \beta, v}^{(l)} D_\tau x^{(l)} - D_{\eta, \beta, v}^{(l)} x^{(l)} \right\| \leq 2^{\beta+1} |\tau|_\infty D_l \left\| x_c^{(l-1)} \right\| \leq 2^{\beta+1} |\tau|_\infty D_l \left\| x^{(0)} \right\|, \quad \forall l \geq 1, \quad (93)$$

where $B_l, C_l, D_l$ are defined in (68) and (69). Lemma 2 thus implies that

$$\left\| x^{(l)}[D_\tau x^{(l-1)}] - D_\tau x^{(l)}[x^{(l-1)}] \right\| \leq 8|\nabla\tau|_\infty \left\| x_c^{(l-1)} \right\| \leq 8|\nabla\tau|_\infty \left\| x^{(0)} \right\|, \quad \forall l \geq 1, \quad (94)$$

$$\left\| D_{\eta, \beta, v}^{(l)} D_\tau x^{(l)} - D_{\eta, \beta, v}^{(l)} x^{(l)} \right\| \leq 2^{\beta+1-j_l} |\tau|_\infty \left\| x_c^{(l-1)} \right\| \leq 2^{\beta+1-j_l} |\tau|_\infty \left\| x^{(0)} \right\|, \quad \forall l \geq 1. \quad (95)$$

For part (b), given any $(\eta, \beta, v) \in S^1 \times \mathbb{R} \times \mathbb{R}^2$, we have

$$\|D_{\eta, \beta, v}^{(l)} x^{(l)}\|^2 = \sup_\alpha \frac{1}{M_l} \sum_\lambda \int_{S^1} \int_{\mathbb{R}^2} \left| D_{\eta, \beta, v}^{(l)} x^{(l)}(u, \theta, \alpha, \lambda) \right|^2 du d\theta \quad (96)$$

$$= \sup_\alpha \frac{1}{M_l} \sum_\lambda \int_{S^1} \int_{\mathbb{R}^2} \left| x^{(l)} (2^{-\beta} R_{-\theta}(u - v), \theta - \eta, \alpha - \beta, \lambda) \right|^2 du d\theta \quad (97)$$

$$= \sup_\alpha \frac{1}{M_l} \sum_\lambda \int_{S^1} \int_{\mathbb{R}^2} \left| x^{(l)}(u', \theta - \eta, \alpha - \beta, \lambda) \right|^2 2^{2\beta} du' d\theta \quad (98)$$

$$= 2^{2\beta} \|x^{(l)}\|^2 \quad (99)$$

Thus (57) is valid. The second half of part (b) holds since

$$\left\| x^{(l)}[D_{\eta,\beta,v}^{(l)} \circ D_\tau x^{(l-1)}] - D_{\eta,\beta,v}^{(l)} D_\tau x^{(l)}[x^{(l-1)}] \right\|$$

$$= \left\| D_{\eta,\beta,v}^{(l)} x^{(l)}[D_\tau x^{(l-1)}] - D_{\eta,\beta,v}^{(l)} D_\tau x^{(l)}[x^{(l-1)}] \right\| \tag{100}$$

$$= 2^\beta \left\| x^{(l)}[D_\tau x^{(l-1)}] - D_\tau x^{(l)}[x^{(l-1)}] \right\| \tag{101}$$

$$\leq 2^{\beta+3} |\nabla \tau|_\infty \|x_c^{(l-1)}\|, \tag{102}$$

where the first equality holds because of the $\mathcal{RST}$-equivariance, i.e., Theorem 1, and the second equality follows from (57).

The proof of part (c) is exactly the same as that of Proposition 3(c) of (Zhu et al., 2019). Specifically, we telescope the inequality (58) while making use of the non-expansiveness of the layer-wise features, i.e., Proposition 2(c). The detail is omitted. □

### C.3 Proof of Theorem 1

*Proof.* Theorem 1 is a direct consequence of Proposition 3. More specifically,

$$\left\| x^{(L)}[D_{\eta,\beta,v}^{(0)} \circ D_\tau x^{(0)}] - D_{\eta,\beta,v}^{(L)} x^{(L)}[x^{(0)}] \right\|$$

$$\leq \left\| x^{(L)}[D_{\eta,\beta,v}^{(0)} \circ D_\tau x^{(0)}] - D_{\eta,\beta,v}^{(L)} D_\tau x^{(L)}[x^{(0)}] \right\| + \left\| D_{\eta,\beta,v}^{(L)} D_\tau x^{(L)}[x^{(0)}] - D_{\eta,\beta,v}^{(L)} x^{(L)}[x^{(0)}] \right\| \tag{103}$$

$$\leq 2^{\beta+3} L |\nabla \tau|_\infty \|x^{(0)}\| + 2^{\beta+1-j_L} |\tau|_\infty \|x^{(0)}\| \tag{104}$$

$$= 2^{\beta+1} \left( 4L|\nabla \tau|_\infty + 2^{-j_L}|\tau|_\infty \right) \|x^{(0)}\|, \tag{105}$$

where the second inequality comes from Proposition 3(c) and Proposition 3(d). This concludes the proof of Theorem 1. □

## D Reproducibility

### D.1 License of Datasets

- **MNIST.** Creative Commons Attribution-Share Alike 3.0 license LeCun (1998).

- **Fashion-MNIST.** MIT license Xiao et al. (2017b).

- **STL-10.** BSD 3-Clause License Coates et al. (2011a).

### D.2 Code Implementation

Our code and experiments in our paper are available at https://github.com/gaoliyao/Roto-scale-translation-Equivariant-CNN. We specifically include the experiments for MNIST and Fashion-MNIST for this version.

The code is built upon the GitHub repository of the paper Sosnovik et al. (2020) under MIT license https://github.com/ISosnovik/sesn. For the implementation of Fourier-Bessel bases and Decomposed Convolutional Filters, we build our code on Zhu et al. (2019). In the submitted files, we keep the names of the authors of Sosnovik et al. (2020), while our names remain anonymous throughout the repository.

