# OpenReview forum: "Deformation Robust Roto-Scale-Translation Equivariant CNNs"
_TMLR — Accepted by TMLR_

### Review · Reviewer_3Kxh · 2022-04-23

**Summary Of Contributions:**

The paper makes three contributions:
1. The first CNN to simultaneously achieve equivariance to rotation, scale, and translation
2. An analysis of the stability to deformations outside of this group, and a method of constraining the filters to achieve stability.
3. Results on in distribution and out of distribution performance on small image datasets

Achieving simultaneous equivariance to rotation, scale and translation is quite difficult and has not been achieved before as far as I know. And most papers on equivariance only consider the transformations inside the group of interest, and not general deformations (diffeomorphisms) around it. Being exactly equivariant to deformations is not feasible, but as the paper shows, one can make the network stable to them.

**Broader Impact Concerns:**

No concerns beyond those that apply to all work in DL architectures.

**Requested Changes:**

Typo in Section 6, bullet 1: 'realist' -> 'realistic'

**Strengths And Weaknesses:**

The paper is well written, and the mathematical background material on groups, actions, representations and equivariance is explained in a clear, concise and correct way.

As mentioned, the paper does something new that is quite hard (RST equivariance).

The stability analysis is a somewhat unique feature and is fairly convincing. The assumptions seem quite reasonable to me, and the resulting method is not too complicated to implement (requires a filter expansion as most equivariant CNNs).

The main weakness is in the scalability of the proposed method, and the small scale experiments. The paper uses regular representations and associated to that convolutions on the group itself. Since the group is 4 dimensional, this means feature maps are 4 dimensional. Aside from the rest of the implementation details, which seem not too problematic, the fact that you need to convolve in four dimensions makes the method compute and memory intensive. A better approach, as the authors note, would be to develop a steerable CNN version of the method, so that only 2D convolutions are required, and this is left for future work.

The other weakness of the paper is that the method is only evaluated on small and simple datasets. It would be nice to see future work (with a more scalable method) evaluated on tasks where scale equivariance is critical, such as perhaps ego-centric vision datasets.

---

### Review · Reviewer_iYwU · 2022-05-02

**Summary Of Contributions:**

The paper proposes new architectures satisfying equivariance to the roto-scale-translation group, using (truncated) decompositions of convolutional filters in appropriate bases. The authors then show deformation stability bounds, showing that when appropriate norms of the filter coefficients are controlled, the model is near-equivariant to roto-scale-translations even when the input is slightly deformed.
The authors then provide numerical experiments on image datasets to illustrate the benefits of these architectures.

**Requested Changes:**

presentation:
* it is often hard to distinguish text and citations, e.g. in section 2. Please use \citep more abundantly.
* many equation references point to the appendix instead of the main text, please fix this

a couple of points that should be discussed more thoroughly:
* the meaning of norms in assumption (A2): it is stated that these norms control certain smoothness properties of filters which help with stability -- it would be great to expand on this further
* regarding scale invariance, you mention that using a single value of alpha, i.e. a single scale ($L_\alpha = 1$ with $\alpha = 0$ only) is best. I'm not sure I understand how such a model provides any scale invariance, given that you're only looking at a single scale?

**Strengths And Weaknesses:**

Strenghts: the work leads to new practical architectures for incorporating roto-scale invariances in CNNs while ensuring deformation stability by truncating filter decompositions to low-frequency components in appropriate bases. Empirically, these perform well.

Weaknesses: the paper is quite incremental, as it mostly builds upon the previous works by Qiu et al (2018), Cheng et al (2019), Zhu et al (2019), which studied very similar constructions for translation, rotation and scaling, separately (rather than jointly).

---

### Review · Reviewer_3L9Y · 2022-05-25

**Summary Of Contributions:**

The primary contribution of the paper is to present a roto-scale-translation equivariant CNN for vision tasks. The paper begins with the motivation that these aspects are usually treated separately in NN design, but in many cases (such as in vision systems of autonomous vehicles), they are best treated jointly. The underlying mathematics is presented in a clear and concise manner, including the necessity and sufficiency of the correct notion of convolution (following now standard results in the literature). A secondary contribution is to present a (deformation) stability analysis (when the underlying "transformation" is not perfect  -- as might be seen in many cases of interest). Experiments on three tasks (one being a standard benchmark) are used to verify the efficacy of the approach.

**Broader Impact Concerns:**

No impact concerns.

**Requested Changes:**

See the last two points made in "weaknesses."

**Strengths And Weaknesses:**

Strengths:
- The idea is quite clean and straightforward.
- The paper is written very clearly, with all the appropriate mathematics introduced in an understandable manner. Overall, the paper is a pleasure to read.
- The claimed contribution of the paper (of treating scale + rotations + translations together) is indeed novel (although some papers, like the Polar transformer networks of Esteves et al. come close).

Despite liking the paper and thinking it has a clear and useful contribution, in my opinion, it also has the following weaknesses:
- The implementation details indicate that the RST architecture uses a truncated interval and expands filters of the G-CNN in the scale-dimension. This, I am afraid, suggests that it is a straightforward extension of the usual GCNN implementation-wise.
- The method has a strong reliance on two papers (both cited in the paper): Cheng et al. and Sosnovik et al. With the former the main difference is the introduction of another dimension to handle scale. The second introduced steerable scale-equivariant filters. The paper is essentially a combination of these two works. The use of truncated intervals is present in most steerable CNNs (since one has to choose a finite basis).
- The above can be overlooked if the experimental evaluation is scaled up and made more convincing. While the experiments are thorough on the datasets that are considered, they do need to be matched with other similar works in the area. Another source of weakness is that some modifications are made to existing baselines -- I think they should be compared to published results. I also think that the most appropriate datasets would be SIM2MNIST (from Esteves et al. "Polar Transformer Networks" ICLR'18) and MNIST-RTS (from Jaderberg et al, "Spatial Transformer Networks", NeurIPS'15). The one benchmark used is STL-10. However, the results reported are far from the state-of-art (including those for published GCNN baselines). I think this weakens the paper.
- The deformation stability contribution needs to be better integrated into the wider paper. It is a bit abstract -- it would strengthen the paper if more examples and experiments are presented that validate their motivations and result.

---

### Decision · Action_Editors · 2022-07-04

**Recommendation:** Accept as is

**Comment:**

This paper describes a CNN that achieves simultaneous invariance to rotations, scales, and translations. Reviewers noted that the problem is challenging, the approach is clean, and the paper is well-written, but they initially raised concerns regarding the empirical comparison between the proposed approach and previous work. The authors addressed these concerns with additional experiments during the rebuttal period. Reviewers also noted that the paper is essentially a combination of approaches from previous work, but given the difficulty of the problem, the AE believes that the paper is likely to be of interest to some individuals in TMLR's audience. All reviewers now either learn toward or recommend accepting the paper. The AE is thus happy to recommend accepting the paper as is.

---

> ### Author Response · Authors · 2022-07-06
> **Thank you!**
>
> We wish to appreciate all the reviewers and the action editor for the effort in reviewing our manuscript. The de-anonymised version with an official Github repo link will be soon uploaded!
>
> Sincerely,
>
> TMLR Paper 29 Authors